# THE DYNAMIC OF CONSENSUS IN DEEP NETWORKS AND THE IDENTIFICATION OF NOISY LABELS

## ABSTRACT

Deep neural networks have incredible capacity and expressibility, and can seemingly memorize any training set. This introduces a problem when training in the presence of noisy labels, as the noisy examples cannot be distinguished from clean examples by the end of training. Recent research has dealt with this challenge by utilizing the fact that deep networks seem to memorize clean examples much earlier than noisy examples. Here we report a new empirical result: for each example, when looking at the time it has been memorized by each model in an ensemble of networks, the diversity seen in noisy examples is much larger than the clean examples. We use this observation to develop a new method for noisy labels filtration. The method is based on a statistics of the data, which captures the differences in ensemble learning dynamics between clean and noisy data. We test our method on three tasks: (i) noise amount estimation; (ii) noise filtration; (iii) supervised classification. We show that our method improves over existing baselines in all three tasks using a variety of datasets, noise models, and noise levels. Aside from its improved performance, our method has two other advantages. (i) Simplicity, which implies that no additional hyperparameters are introduced. (ii) Our method is modular: it does not work in an end-to-end fashion, and can therefore be used to clean a dataset for any other future usage.

## 1 INTRODUCTION

Deep neural networks dominate the state of the art in an ever increasing list of application domains, but for the most part, this incredible success relies on very large datasets of annotated examples available for training. Unfortunately, large amounts of high-quality annotated data are hard and expensive to acquire, whereas cheap alternatives (obtained by way of crowd-sourcing or automatic labeling, for example) often introduce noisy labels into the training set. By now there is much empirical evidence that neural networks can memorize almost every training set, including ones with noisy and even random labels (Zhang et al., 2017), which in turn increases the generalization error of the model. As a result, the problems of identifying the existence of label noise and the separation of noisy labels from clean ones, are becoming more urgent and therefore attract increasing attention.

Henceforth, we will call the set of examples in the training data whose labels are correct "clean data", and the set of examples whose labels are incorrect "noisy data". While all labels can be eventually learned by deep models, it has been empirically shown that most noisy datapoints are learned by deep models late, after most of the clean data has already been learned (Arpit et al., 2017). Therefore many methods focus on the learning time of an example in order to classify it as noisy or clean, by looking at its loss (Pleiss et al., 2020; Arazo et al., 2019) or loss per epoch (Li et al., 2020) in a single model. However, these methods struggle to classify correctly clean and noisy datapoints that are learned at the same time, or worse - noisy datapoints that are learned early. Additionally, many of these methods work in an end-to-end manner, and thus neither provide noise level estimation nor do they deliver separate sets of clean and noisy data for novel future usages.

Our first contribution is a new empirical results regarding the learning dynamics of an ensemble of deep networks, showing that the dynamics is different when training with clean data vs. noisy data. The dynamics of clean data has been studied in (Hacohen et al., 2020; Pliushch et al., 2021), where it is reported that different deep models learn examples in the same order and pace. This means that when training a few models and comparing their predictions, a binary occurrence (approximately)

is seen at each epoch $e$: either all the networks correctly predict the example's label, or none of them does. This further implies that for the most part, the distribution of predictions across points is bimodal. Additionally, a variety of studies showed that the bias and variance of deep networks decrease as the networks complexity grow (Nakkiran et al., 2021; Neal et al., 2018), providing additional evidence that different deep networks learn data at the same time simultaneously.

In Section 3 we describe a new empirical result: when training an ensemble of deep models with noisy data, and *in contrast to what happens when using clean data, different models learn different datapoints at different times* (see Fig. 1). This empirical finding tells us that in an ensemble of networks, the learning dynamics of clean data and noisy data can be distinguished. When training such an ensemble with a mixture of clean and noisy data, the emerging dynamics reflects this observation, as well as the tendency of clean data to be learned faster as previously observed.

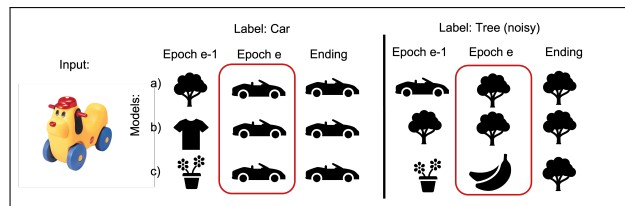

Figure 1: With noisy labels models show higher disagreement. The noisy examples are not only learned at a later stage, but each model learns the example at its own different time.

In our second contribution, we use this result to develop a new algorithm for noise level estimation and noise filtration, which we call *DisagreeNet* (see Section 4). Importantly, unlike most alternative methods, our algorithm is simple (it does not introduce any new hyperparameters), parallelizable, easy to integrate with any supervised or semi-supervised learning method and any loss function, and does not rely on prior knowledge of the noise amount. When used for noise filtration, our empirical study (see Section 5) shows the superiority of *DisagreeNet* as compared to the state of the art, using different datasets, different noise models and different noise levels. When used for supervised classification by way of pre-processing the training set prior to training a deep model, it provides a significant boost in performance, more so than alternative methods.

**Relation to prior art**

Work on the dynamics of learning in deep models has received increased attention in recent years (e.g., Nguyen et al., 2020; Hacohen et al., 2020; Baldock et al., 2021). Our work adds a new observation to this body of knowledge, which is seemingly unique to an ensemble of deep models (as against an ensemble of other commonly used classifiers). Thus, while there exist other methods that use ensembles to handle label noise (e.g., Sabzevari et al., 2018; Feng et al., 2020; Chai et al., 2021; de Moura et al., 2018), for the most part they cannot take advantage of this characteristic of deep models, and as a result are forced to use additional knowledge, typically the availability of a clean validation set and/or prior knowledge of the noise amount.

Work on deep learning with noisy labels (see Song et al. (2022) for a recent survey) can be coarsely divided to two categories: general methods that use a modified loss or network's architecture, and methods that focus on noise identification. The first group includes methods that aim to estimate the underlying noise transition matrix (Goldberger and Ben-Reuven, 2016; Patrini et al., 2017), employ a noise-robust loss (Ghosh et al., 2017; Zhang and Sabuncu, 2018; Wang et al., 2019; Xu et al., 2019), or achieve robustness to noise by way of regularization (Tanno et al., 2019; Jenni and Favaro, 2018). Methods in the second group, which is more inline with our approach, focus more directly on noise identification. Some methods assume that clean examples are usually learned faster than noisy examples (e.g. Liu et al., 2020). Others (Arazo et al., 2019; Li et al., 2020) generate soft labels by interpolating the given labels and the model's predictions during training. Yet other methods (Jiang et al., 2018; Han et al., 2018; Malach and Shalev-Shwartz, 2017; Yu et al., 2019; Lee and Chung, 2019), like our own, inspect an ensemble of networks, usually in order to transfer information between networks and thus avoid agreement bias.

Notably, we also analyze the behavior of ensembles in order to identify the noisy examples, resembling (Pleiss et al., 2020; Nguyen et al., 2019; Lee and Chung, 2019). But unlike these methods, which track the loss of the networks, we track the dynamics of the agreement between multiple networks over epochs. We then show that this statistic is more effective, and achieves superior results. Additionally (and not less importantly), unlike these works, we do not assume prior knowledge of the noise amount or the presence of a clean validation set, and do not introduce new hyper-parameters in our algorithm.

Recently, the emphasis has somewhat shifted to the use of semi-supervised learning and contrastive learning (Li et al., 2020; Liu et al., 2020; Ortego et al., 2020; Wei et al., 2020; Yao et al., 2021; Zheltonozhskii et al., 2022; Li et al., 2022; Karim et al., 2022). Semi-supervised learning is an effective paradigm for the prediction of missing labels. This paradigm is especially useful when the identification of noisy points cannot be done reliably, in which case it is advantageous to remove labels whose likelihood to be true is not negligible. The effectiveness of semi-supervised learning in providing reliable pseudo-labels for unlabeled points will compensate for the loss of clean labels.

However, semi-supervised learning is not universally practical as it often relies on the extraction of effective representations based on unsupervised learning tasks, which typically introduces implicit priors (e.g., that contrastive loss is appropriate). In contrast, our goal is to reliably identify noisy points, to be subsequently removed. Thus, our method can be easily incorporated into any SOTA method which uses supervised or semi-supervised learning (with or without contrastive learning), and may provide benefit even when semi-supervised learning is not viable.

## 2    INTER-NETWORK AGREEMENT: DEFINITION AND SCORES

Measuring the similarity between deep models is not a trivial challenge, as modern deep neural networks are complex functions defined by a huge number of parameters, which are invariant to transformations hidden in the model's architecture. Here we measure the similarity between deep models in an ensemble by measuring inter-model prediction agreement at each datapoint. Accordingly, in Section 2.2 we describe scores that are based on the state of the networks at each epoch $e$, while in Section 2.3 we describe cumulative scores that integrate these states through many epochs. Practically (see Section 4), our proposed method relies on the cumulative scores, which are shown empirically to provide more accurate results in the noise filtration task. These scores promise added robustness, as it is no longer necessary to identify the epoch at which the score is to be evaluated.

### 2.1    PRELIMINARIES

**Notations** Let $f^e : \mathbb{R}^d \to [0, 1]^{|C|}$ denote a deep model, trained with Stochastic Gradient Descent (SGD) for $e$ epochs on training set $\mathbb{X} = \{(\boldsymbol{x}_i, y_i)\}_{i=1}^{M}$, where $\boldsymbol{x}_i \in \mathbb{R}^d$ denotes a single example and $y_i \in [C]$ its corresponding label. Let $\mathcal{F}^e(\mathbb{X}) = \{f_1^e, ..., f_N^e\}$ denote an ensemble of $N$ such models, where each model $f_{i \in [N]}^e$ is initialized and trained independently on $\mathbb{X}$ for $\mathcal{E}$ epochs.

**Noise model** We analyze the training dynamics of an ensemble of models in the presence of label noise. Label noise is different from data noise (like image distortion or additive Gaussian noise). Here it is assumed that after the training set $\mathbb{X} = \{(\boldsymbol{x}_i, l_i)\}_{i=1}^{M}$ is sampled, the labels $\{l_i\}$ are corrupted by some noise function $g : [C] \to [C]$, and the training set becomes $\mathbb{X} = \{(\boldsymbol{x}_i, y_i)\}_{i=1}^{M}$, $y_i = g(l_i)$. The two most common models of label noise are termed *symmetric noise* and *asymmetric noise* (Patrini et al., 2017). In both cases it is assumed that some fixed percentage of the labels are corrupted by $g(l)$. With symmetric noise, $g(l)$ assigns any new label from the set $[C] \setminus \{l\}$ with equal probability. With asymmetric noise, $g(l)$ is the deterministic permutation function (see App. F for details). Note that the asymmetric noise model is considered much harder than the symmetric noise model.

### 2.2    PER-EPOCH AGREEMENT SCORE

Following Hacohen et al. (2020), we define the *True Positive Agreement* (TPA) score of ensemble $\mathcal{F}^e(\mathbb{X})$ at each datapoint $(\boldsymbol{x}, y)$, where $\boxed{TPA(\boldsymbol{x}, y; \mathcal{F}^e(\mathbb{X})) = \frac{1}{N} \sum_{i=1}^{N} \mathbb{1}_{[f_i^e(\boldsymbol{x}) = y]}}$. The TPA score measures the average accuracy of the models in the ensemble, when seeing $\boldsymbol{x}$, after each model has been trained for exactly $e$ epochs on $\mathbb{X}$. Note that $TPA$ measures the average accuracy of multiple models on one example, as opposed to the generalization error that measures the average error of one model on multiple examples.

### 2.3    CUMULATIVE SCORES

When inspecting the dynamics of the TPA score on clean data, we see that at the beginning the distribution of $\{TPA(\boldsymbol{x}_i, y_i)\}$ is concentrated around 0, and then quickly shifts to 1 as training proceeds (see side panels in Fig. 2a). This implies that empirically, data is learned in a specific order

by all models in the ensemble. To measure this phenomenon we use the *Ensemble Learning Pace* (ELP) score defined below, which essentially integrates the TPA using all training epochs:

$$ELP(\boldsymbol{x}, y) = \frac{1}{\mathcal{E}} \sum_{e \in [1, \ldots, \mathcal{E}]} TPA(\boldsymbol{x}, y; \mathcal{F}^e(\mathbb{X})) \tag{1}$$

$ELP(\boldsymbol{x}, y)$ captures both the time of learning by a single model, and its consistency across models. For example, if all the models learned the example early, the score would be high. It would be significantly lower if some of them learned it later than others (see pseudo-code in App. C).

In our study we evaluated two additional cumulative scores of inter-model agreement:

1. Cumulative loss:
$$CumLoss(\boldsymbol{x}, y) = \frac{1}{N\mathcal{E}} \sum_{i,e \in [1,\ldots,\mathcal{E}]} CE(f_i^e(\boldsymbol{x}), y)$$

   Above $CE$ denotes the cross entropy function. This score is very similar to ELP, engaging the average of the cross-entropy loss instead of the accuracy indicator $\mathbb{1}_{[f_i^e(\boldsymbol{x})=y]}$.

2. Area under the margin: following (Pleiss et al., 2020), the MeanMargin score is defined as follows
$$MeanMargin(\boldsymbol{x}, y) = \frac{1}{N\mathcal{E}} \sum_{i,e \in [1,\ldots,\mathcal{E}]} [f_i^e(\boldsymbol{x})]_{y_i} - \operatorname*{argmax}_{j \neq y_i} [f_i^e(\boldsymbol{x})]_j$$

   The MeanMargin score is the mean of the 'margin', the difference between the value of the ground-truth logit (before softmax) and the value of the otherwise maximal logit.

## 3 THE DYNAMICS OF AGREEMENT: NEW EMPIRICAL OBSERVATION

In this section we analyze, both theoretically and empirically, how measures of inter-network agreement may indicate the detrimental phenomenon of *Overfit*. *Overfit* is a condition that can occur during the training of deep neural networks. It is characterized by the co-occurring decrease of *train error or loss* and the increase of *test error or loss*. Recall that train loss is the quantity that is being continuously minimized during the training of deep models, while the test error is the quantity linked to generalization error. When these quantities change in opposite directions, training harms the final performance and thus early stopping is recommended.

We begin by showing in Section 3.1 that in an ensemble of linear regression models, overfit and the agreement between models are negatively correlated. When this is the case, an epoch in which the agreement between networks reaches its maximal value is likely to indicate the beginning of overfit.

Our next goal is to examine the relevance of this result to deep learning in practice. Yet inexplicably, at least as far as image datasets are concerned, overfit rarely occurs in practice when deep learning is used for image recognition. However, when label noise is introduced, significant overfit occurs. Capitalizing on this observation, we report in Section 3.3 that when overfit occurs in the independent training of an ensemble of deep networks, the agreement between the networks starts to decrease.

The approach we describe in Section 4 is motivated by these results: Since it has been observed that noisy data are memorized later than clean data, we hypothesize that overfit occurs when the memorization of noisy labels becomes dominant. This suggests that measuring the dynamics of agreement between networks, which is correlated with overfit as shown below, can be effectively used for the identification of label noise.

### 3.1 OVERFIT AND AGREEMENT: THEORETICAL RESULT

Since deep learning models are not amenable to a rigorous theoretical analysis, and in order to gain computational insight into such general phenomena as overfit, simpler models are sometimes analyzed (e.g. Weinshall and Amir, 2020). Accordingly, in App. A we analyze the relation between overfit and inter-model agreement in an ensemble of linear regression models. Our analysis culminates in a theorem, which states that the agreement between linear regression models decreases when overfit occurs in all the models, namely, when the generalization error in all the models increases. Here is a brief sketch of the theorem's proof (see Appendix A):

- *Disagreement* is measured by the empirical variance over models of the error vector at each test point, averaged over the test examples.

- We prove the intuitive Lemma 1, stating the following: overfit occurs in a model iff the gradient step of the model, which is computed from the training set, is negatively correlated with a vector unknown to the learner - the gradient step defined by the test set.

- Using some asymptotic assumptions and a lengthy and technical derivation, we show that the disagreement is approximately the sum of the correlations between each network's gradient step and its "test gradient step". Then, it follows immediately from Lemma 1 that if overfit occurs in all the models then the aforementioned disagreement score increases.

## 3.2 MEASURING THE AGREEMENT BETWEEN MODELS

In order to obtain a score that captures the level of disagreement between networks, we inspect more closely the distribution of $TPA(\boldsymbol{x}, y; \mathcal{F}^e(\mathbb{X}))$, defined in Section 2.2, over a sample of datapoints, and analyze its dynamics as training proceeds. First, note that if all of the models in ensemble $\mathcal{F}^e(\mathbb{X})$ give identical predictions at each point, the TPA score would be either 0 (when all the networks predict a false label) or 1 (when all the networks predict the correct label). In this case, the TPA distribution is perfectly bimodal, with only two peaks at 0 and 1. If the predictions of the models at each point are independent with mean accuracy $p$, then it can be readily shown that TPA is approximately the binomial random variable with a unimodal distribution around $p$.

Empirically, (Hacohen et al., 2020) showed that in ensembles of deep models trained on 'real' datasets as we use here, the TPA distribution is highly bimodal. Since commonly used measures of bimodality, such as the Pearson bimodality score, are ill-fitted for the discrete TPA distribution, we measure bimodality with the following *Bimodal Index* score:

$$BI(e) = \sqrt{\frac{1}{M}\sum_{i=1}^{M}\mathbb{1}_{[TPA(\boldsymbol{x}_i, y_i; \mathcal{F}^e(\mathbb{X}))=N]}} + \sqrt{\frac{1}{M}\sum_{i=1}^{M}\mathbb{1}_{[TPA(\boldsymbol{x}_i, y_i; \mathcal{F}^e(\mathbb{X}))=0]}} \quad (2)$$

$BI(e)$ measures how many examples are either correctly or incorrectly classified by *all* the models in the ensemble, rewarding distributions where points are (roughly) equally divided between 0 and 1. Here we use this score to measure the agreement between networks at epoch $e$.

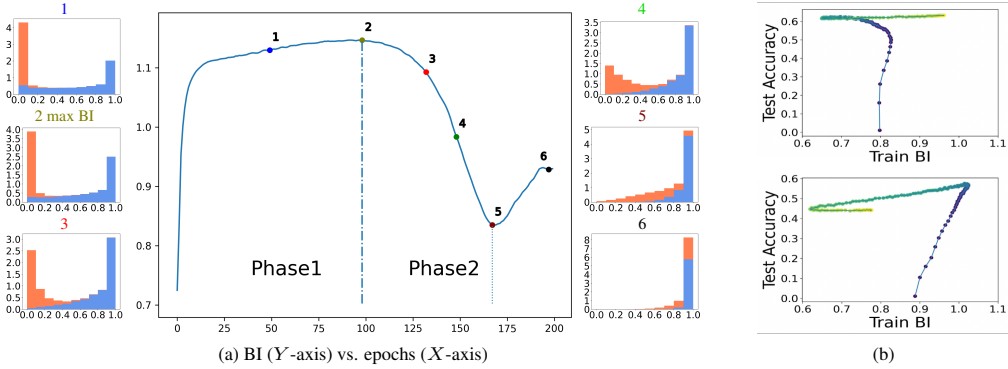

(a) BI ($Y$-axis) vs. epochs ($X$-axis)             (b)

Figure 2: (a) Main panel: bimodality in an ensemble of 10 DenseNet networks, trained to classify Cifar10 with 20% symmetric noise. Side panels: TPA distribution in 6 epochs (blue - clean examples, orange - noisy ones). (b) Scatter plots of test accuracy vs train bimodality, measured by $BI(e)$ as defined in (2), where changes in color from blue to yellow correspond with advancing epochs.

If we were to draw the *Bimodality Index (BI)* of the TPA score as a function of the epochs (Fig. 2a), we often see two distinct phases. Initially (phase 1), BI is monotonically increasing, namely, both test accuracy and agreement are on the rise. We call it the 'learning' phase. Empirically, in this phase most of the clean examples are being learned (or memorized), as can also be seen in the left side panels of Fig. 2a (cf. Li et al., 2015). At some point BI may start to decrease, followed by another possible ascent. This is phase 2, in which empirically the memorization of noisy examples dominates the learning (see the right side panels of Fig. 2a). This fall and rise is explained by another set of empirical observations, that noisy labels are **not** being learned in the same order by an ensemble of networks (see App. B), which therefore predicts a decline in BI when noisy labels are being learned.

### 3.3 OVERFIT AND AGREEMENT: EMPIRICAL EVIDENCE

Earlier work, investigating the dynamics of learning in deep networks, suggests that examples with noisy labels are learned later (Krueger et al., 2017; Zhang et al., 2017; Arpit et al., 2017; Arora et al., 2019). Since the learning of noisy labels is unlikely to improve the model's test accuracy, we hypothesize that this may be correlated with the occurrence (or increase) of *overfit*. The theoretical result in Section 3.1 suggests that this may be correlated with a decrease in the agreement between networks. Our goal now is to test this prediction empirically.

We next outline empirical evidence that this is indeed the case in actual deep models. In order to boost the strength of overfit, we adopt the scenario of recognition with label noise, where the occurrence of overfit is abundant. When overfit indeed occurs, our experiments show that if the test accuracy drops, then the disagreement score BI also decreases (see example in Fig. 2b-bottom). This observation is confirmed with various noise models and different datasets. When overfit does not occur, the prediction is no longer observed (see example in Fig. 2b-top).

These results suggest that a consistent drop in the $BI$ index of some training set $\mathbb{X}$ can be used to estimate the occurrence of overfit, and possibly even the beginning of noisy label memorization.

## 4 DEALING WITH NOISY LABELS: PROPOSED APPROACH

When dealing with noisy labels, there are essentially three intertwined problems that may require separate treatment:

1. **Noise level estimation**: estimate the number of noisy examples.
2. **Noise filtration**: flag points whose label is to be removed.
3. **Classifier construction**: train a model without the examples that are flagged as noisy.

### 4.1 DISAGREENET, FOR NOISE LEVEL ESTIMATION AND NOISE FILTRATION

Guided by Section 3, we propose a method to estimate the noise level in a training set denoted *DisagreeNet*, which is further used to filter out the noisy examples (see pseudo-code below in Alg. 1):

1. Compute the ELP score from (1) at each training example.
2. Fit a two component BMM to the ELP distribution (see Fig. 3).
3. Use the intersection between the 2 components of the BMM fit to divide the data to two groups.
4. Call the group with lower ELP 'noisy data'.
5. Estimate noise level by counting the number of datapoints in the noisy group.

---

**Algorithm 1:** *DisagreeNet*

---

**Input:** ELP_arr, specifying the ELP score of each point in training set $\mathbb{X}$
**Output:** Noise level estimate, and the list of indices of noisy points
$\{G_{low-ELP}, G_{high-ELP}\} \leftarrow$ divide the data to two groups using **fit_BMM**(ELP_arr);
noise_indices $\leftarrow$ indices of ELP_arr assigned to $G_{low-ELP}$;
noise_estim $\leftarrow \frac{|G_{low-ELP}|}{|\text{ELP\_arr}|}$;
**return** noise_estim, noise_indices

---

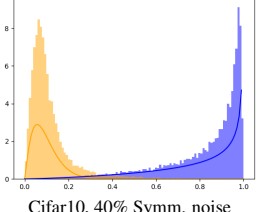 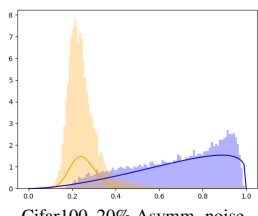 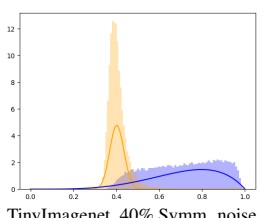

| Cifar10, 40% Symm. noise | Cifar100, 20% Asymm. noise | TinyImagenet, 40% Symm. noise |

Figure 3: ELP distribution, shown separately for the clean data in blue and the noisy data in orange. Superimposed, in blue and orange lines, is the bi-modal BMM fit to the ELP total (not separated) distribution

## 4.2 CLASSIFIER CONSTRUCTION

Aiming to achieve modular handling of noisy labels, we propose the following two-step approach:

1. Run *DisagreeNet*.
2. Run SOTA supervised learning method using the filtered data.

In step 2 it is possible to invoke semi-supervised SOTA methods, using the noisy group as unsupervised data. However, given that semi-supervised learning typically involves additional assumptions (or prior knowledge) as well as high computational complexity (that restricts its applicability to smaller datasets), as discussed in Section 1, we do not consider this scenario here.

## 5 EMPIRICAL EVALUATION

We evaluate our method in the following scenarios and tasks:

1. Noise identification (Section 5.2), with two complementary sub-tasks: (i) estimate the noise level in the given dataset; (ii) identify the noisy examples.
2. Supervised classification (Section 5.3), after the removal of the noisy examples.

### 5.1 DATASET AND BASELINES (DETAILS ARE DEFERRED TO APP. F)

**Datasets** We evaluate our method on a few standard image classification datasets, including Cifar10 and Cifar100 (Krizhevsky et al., 2009), Tiny imagenet (Le and Yang, 2015), subsets of Imagenet (Deng et al., 2009), Clothing1M (Xiao et al., 2015) and Animal10N (Song et al., 2019a), see App. F for details. These datasets were used in earlier work to evaluate the success of noise estimation (Pleiss et al., 2020; Arazo et al., 2019; Li et al., 2020; Liu et al., 2020).

**Baselines and comparable methods** We report results with the following supervised learning methods for learning from noisy data: ***DY-BMM*** and ***DY-GMM*** (Arazo et al., 2019), ***INCV*** (Chen et al., 2019), ***AUM*** (Pleiss et al., 2020), ***Bootstrap*** (Reed et al., 2014), ***D2L*** (Ma et al., 2018), ***MentorNet*** (Jiang et al., 2018), ***FINE*** (Kim et al., 2021). We also report the results of two absolute baselines: (i) ***Oracle***, which trains the model on the clean dataset; (ii) ***Random***, which trains the model after the removal of a random fraction of the whole data, equivalent to the noise level.

**Other methods** The following methods use additional prior information, such as a clean validation set or known level of noise: ***Co-teaching*** (Han et al., 2018), ***O2U*** (Huang et al., 2019b), ***LEC*** (Lee and Chung, 2019) and ***SELFIE*** (Song et al., 2019b). Direct comparison does injustice to the previous group of methods, and is therefore deferred to App. H. Another group of methods is excluded from the comparison because they invoke semi-supervised or contrastive learning (e.g., Ortego et al., 2020; Li et al., 2020; Karim et al., 2022; Li et al., 2022; Wei et al., 2020; Yao et al., 2021), which is a different learning paradigm (see discussion of prior art in Section 1).

**Implementation details** We used DenseNet (Iandola et al., 2014), ResNet-18 and ResNet50 (He et al., 2016) when training on CIFAR-10/100 and Tiny imagenet , and ResNet50 for Clothing1M and Animal10N.

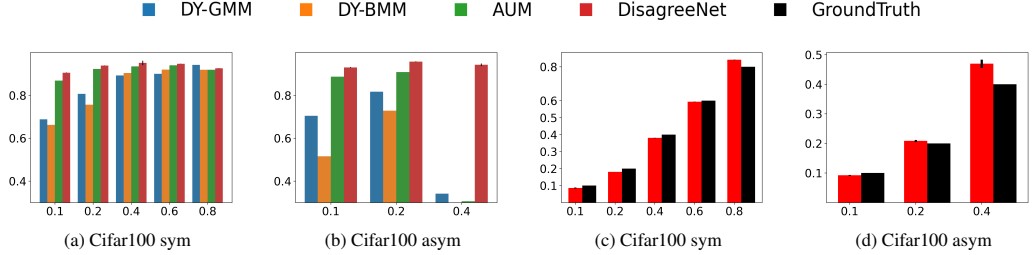

(a) Cifar100 sym    (b) Cifar100 asym    (c) Cifar100 sym    (d) Cifar100 asym

Figure 4: (a)-(b) **Noise identification**: F1 score for noisy label identification task, using different noise levels ($X$-axis), with asymmetric (a) and asymmetric (b) noise models. Results reflect 3 repetitions involving an ensemble of 10 Densenets each. (c)-(d) **Noise level estimation**: different noise levels are evaluated ($X$-axis), with asymmetric (c) and asymmetric (d) noise models (the 3 comparison baselines did not report this estimate).

## 5.2 RESULTS: NOISE IDENTIFICATION

The performance of *DisagreeNet* is evaluated in two tasks: (i) The detection of noisy examples, shown in Fig. 4a-4b (see also Figs. 7 and 8 in App. D), where *DisagreeNet* is seen to outperform the three baselines - *AUM*, *DY-GMM* and *DY-BMM*. (ii) Noise level estimation, shown in Fig. 4c-4d, showing good noise level estimation especially in the case of symmetric noise. We also compare *DisagreeNet* to MeanMargin and CumLoss, see Fig. 5.

## 5.3 RESULT: SUPERVISED CLASSIFICATIONS

Table 1: Test accuracy (%), average and standard error, in the best epoch of retraining after filtration. Results of benchmark methods (see Section 5.1) are taken from (Pleiss et al., 2020) except FINE (Kim et al., 2021), which was re-implemented by us using the official code. The top and middle tables show CIFAR-10, CIFAR-100 and Tiny Imagenet, with simulated noise. The bottom table shows three 'real noise' datasets, and includes in addition results of noise level estimation (when applicable). The presumed noise level for these datasets is indicated in the top line following (Huang et al., 2019a; Song et al., 2019b).

| Method/**Dataset** | **CIFAR-10 sym** | | | **CIFAR-100 sym** | | |
|---|---|---|---|---|---|---|
| Noise level | 20% | 40% | 60% | 20% | 40% | 60% |
| random | $87.18 \pm 0.6$ | $81.59 \pm 0.4$ | $64.35 \pm 0.4$ | $65.49 \pm 0.4$ | $49.1 \pm 0.2$ | $28.7 \pm 0.5$ |
| *Bootstrap* | $77.6 \pm 0.2$ | $62.6 \pm 0.4$ | $48.0 \pm 0.2$ | $51.4 \pm 0.2$ | $41.1 \pm 0.2$ | $29.7 \pm 0.2$ |
| *MentorNet* | $86.7 \pm 0.1$ | $81.9 \pm 0.2$ | – | $64.2 \pm 0.3$ | $57.5 \pm 0.2$ | – |
| *D2L* | $87.7 \pm 0.2$ | $84.4 \pm 0.3$ | $72.7 \pm 0.6$ | $54.0 \pm 1.0$ | $29.7 \pm 1.8$ | – |
| *INCV* | $89.5 \pm 0.1$ | $86.8 \pm 0.1$ | $81.1 \pm 0.3$ | $58.6 \pm 0.5$ | $55.4 \pm 0.2$ | $43.7 \pm 0.3$ |
| *AUM* | $90.2 \pm 0.0$ | $87.5 \pm 0.1$ | $82.1 \pm 0.0$ | $65.5 \pm 0.2$ | $61.3 \pm 0.1$ | $53.0 \pm 0.5$ |
| *FINE* | $92.7$ | $90.9$ | $81.5$ | $73.2$ | $66.8$ | $52.04$ |
| *DisagreeNet*+SL | $\mathbf{93.1 \pm 0.2}$ | $\mathbf{91.1 \pm 0.1}$ | $\mathbf{83.9 \pm 0.08}$ | $\mathbf{77.3 \pm 0.2}$ | $\mathbf{71.8 \pm 0.3}$ | $\mathbf{64.7 \pm 0.3}$ |
| oracle | $95.1 \pm 0.2$ | $94.1 \pm 0.2$ | $92.4 \pm 0.1$ | $78.2 \pm 0.3$ | $75.4 \pm 0.1$ | $70.3 \pm 0.2$ |

| Method/**Dataset** | **CIFAR-10 constant asym** | | **CIFAR-100 constant asym** | | **Tiny Imagenet sym** | |
|---|---|---|---|---|---|---|
| Noise level | 20% | 40% | 20% | 40% | 20% | 40% |
| *random* | $89.5 \pm 0.2$ | $79.3 \pm 0.4$ | $65.2 \pm 0.1$ | $44.64 \pm 0.2$ | $49.8 \pm 0.4$ | $29.9 \pm 0.3$ |
| *Bootstrap* | $76.2 \pm 0.2$ | $55.0 \pm 0.6$ | $53.4 \pm 0.3$ | $38.7 \pm 0.3$ | - | - |
| *D2L* | $88.6 \pm 0.2$ | $76.4 \pm 1.5$ | $43.6 \pm 0.7$ | $16.9 \pm 1.2$ | - | - |
| *DY-BMM* | $77.9 \pm 0.1$ | $59.4 \pm 0.6$ | $53.2 \pm 0.0$ | $37.9 \pm 0.0$ | $41.8 \pm 0.1$ | $36.3 \pm 0.2$ |
| *INCV* | $88.3 \pm 0.1$ | $79.8 \pm 0.4$ | $56.8 \pm 0.1$ | $44.4 \pm 0.7$ | $45.2 \pm 0.1$ | $42.6 \pm 0.1$ |
| *AUM* | $89.7 \pm 0.1$ | $58.7 \pm 0.2$ | $59.7 \pm 0.2$ | $40.2 \pm 0.1$ | $48.9 \pm 0.2$ | $44.7 \pm 0.1$ |
| *FINE* | $92.3$ | $87.4$ | $73.04$ | $61.03$ | $53.8$ | $50.3$ |
| *DisagreeNet*+SL | $\mathbf{94.4 \pm 0.1}$ | $\mathbf{91.9 \pm 0.0}$ | $\mathbf{73.9 \pm 0.5}$ | $\mathbf{61.3 \pm 0.2}$ | $\mathbf{64.5 \pm 0.1}$ | $\mathbf{58.5 \pm 0.2}$ |
| oracle | $95.2 \pm 0.0$ | $94.3 \pm 0.0$ | $78.1 \pm 0.1$ | $75 \pm 0.1$ | $65.4 \pm 0.0$ | $60.8 \pm 0.2$ |

| Method/**Dataset** | **animal10N, 8% noise** | | **Clothing1M, 38% noise** | |
|---|---|---|---|---|
| Noise level | noise est | test accuracy | noise est | test accuracy |
| *Cross-Entropy* | - | $84.1 \pm 0.3$ | - | 69 |
| *AUM* | - | - | 10.7 | 70.4 |
| *DisagreeNet*+SL | 7.8 | $85.1 \pm 0.1$ | 17 | 70.8 |

*DisagreeNet* is used to remove noisy examples, after which we train a deep model from scratch using the remaining examples only. We report our main results using the Densenet architecture, and report results with other architectures in the ablation study. Table 1 summarizes the results for simulated symmetric and asymmetric noise on 5 datasets, and 3 repetitions. It also shows results on 2 real datasets, which are assumed (in previous work) to contain significant levels of 'real' label noise. Additional results are reported in App. H, including methods that require additional prior knowledge.

Not surprisingly, dealing with datasets that are presumed to include inherent label noise proved more difficult, and quite different, than dealing with synthetic noise. As claimed in (Ortego et al., 2020), non-malicious label noise does less damage to networks' generalization than random label noise: on Clothing1M, for example, hardly any overfit is seen during training, even though the data is believed to contain more than 35% noise. Still, here too, *DisagreeNet* achieves improved accuracy without access to a clean validation set or known noise level (see Table 1). In App. H, Table 8 we compare *Disagreenet* to methods that *do* use such prior knowledge. Surprisingly, we see that *DisagreeNet* still achieves better results even without using any additional prior knowledge.

### 5.4 ABLATION STUDY

**How many networks are needed?** We report in Table. 2 the F1 score for noisy label identification, using *DisagreeNet* with varying numbers of networks. The main boost in performance provided by the use of additional networks is seen when using *DisagreeNet* on hard noise scenarios, such as the asymmetric noise, or with small amounts of noise.

Table 2: F1 score of DisagreeNet, using different numbers of models.

| Dataset | Noise | size of ensemble (number of networks) | | | | | |
|---|---|---|---|---|---|---|---|
| Method | | 1 | 2 | 3 | 4 | 7 | 10 |
| Cifar10 sym | 10% | $0.605 \pm 0.01$ | $0.77 \pm 0.0$ | $0.862 \pm 0.0$ | $0.906 \pm 0.0$ | $0.941 \pm 0.0$ | $0.936 \pm 0.0$ |
| | 20% | $0.861 \pm 0.0$ | $0.939 \pm 0.0$ | $0.95 \pm 0.0$ | $0.949 \pm 0.0$ | $0.943 \pm 0.0$ | $0.941 \pm 0.0$ |
| | 40% | $0.954 \pm 0.0$ | $0.953 \pm 0.0$ | $0.952 \pm 0.0$ | $0.952 \pm 0.0$ | $0.951 \pm 0.0$ | $0.951 \pm 0.0$ |
| Cifar100 sym | 10% | $0.225 \pm 0.05$ | $0.855 \pm 0.0$ | $0.855 \pm 0.01$ | $0.854 \pm 0.0$ | $0.860 \pm 0.01$ | $0.864 \pm 0.01$ |
| | 20% | $0.89 \pm 0.0$ | $0.895 \pm 0.0$ | $0.896 \pm 0.0$ | $0.897 \pm 0.0$ | $0.901 \pm 0.0$ | $0.899 \pm 0.0$ |
| | 40% | $0.89 \pm 0.0$ | $0.917 \pm 0.0$ | $0.921 \pm 0.0$ | $0.924 \pm 0.0$ | $0.924 \pm 0.0$ | $0.927 \pm 0.0$ |
| Cifar10 asym | 10% | $0.355 \pm 0.0$ | $0.469 \pm 0.01$ | $0.568 \pm 0.01$ | $0.631 \pm 0.01$ | $0.748 \pm 0.0$ | $0.814 \pm 0.0$ |
| | 20% | $0.553 \pm 0.0$ | $0.642 \pm 0.01$ | $0.703 \pm 0.01$ | $0.734 \pm 0.0$ | $0.799 \pm 0.01$ | $0.829 \pm 0.01$ |
| | 40% | $0.739 \pm 0.0$ | $0.795 \pm 0.0$ | $0.816 \pm 0.0$ | $0.826 \pm 0.0$ | $0.824 \pm 0.0$ | $0.812 \pm 0.0$ |
| Cifar100 asym | 10% | $0.703 \pm 0.0$ | $0.708 \pm 0.0$ | $0.712 \pm 0.0$ | $0.716 \pm 0.0$ | $0.718 \pm 0.0$ | $0.717 \pm 0.0$ |
| | 20% | $0.727 \pm 0.0$ | $0.732 \pm 0.0$ | $0.732 \pm 0.0$ | $0.735 \pm 0.0$ | $0.736 \pm 0.0$ | $0.737 \pm 0.0$ |
| | 40% | $0.594 \pm 0.0$ | $0.606 \pm 0.0$ | $0.614 \pm 0.0$ | $0.614 \pm 0.0$ | $0.618 \pm 0.0$ | $0.62 \pm 0.0$ |

**Additional ablation results** Results in App. E, Table 3 indicate robustness to architecture, scheduler, and usage of augmentation, although the standard training procedures achieve the best results. Additionally, we see robustness to changing the backbone architecture of *DisagreeNet*, using ResNet18 and ResNet50, see App. E, Table 6. Finally, in Fig. 5 we compare *DisagreeNet* using ELP to disagreeNet using the MeanMargin and CumLoss scores, as defined in Section 2.3. In symmetric noise scenarios all scores perform well, while in asymmetric noise scenarios the ELP score performs much better, as can be seen in Figs. 5b,5d. Additional comparisons of the 3 scores are reported in Apps. E and G.

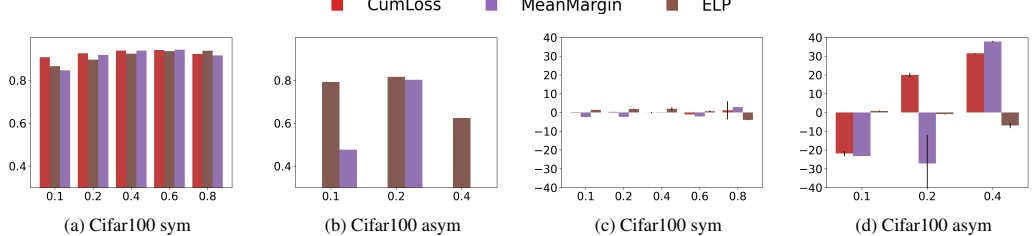

(a) Cifar100 sym    (b) Cifar100 asym    (c) Cifar100 sym    (d) Cifar100 asym

Figure 5: (a)-(b): F1 score ($Y$-axis) for the noisy label identification task, using different noise levels ($X$-axis), with asymmetric (a) and asymmetric (b) noise models. Results with 3 variants of *DisagreeNet* are shown, based on 3 scores: MeanMargin, ELP and CumLoss. (c)-(d): Error in noise level estimation ($Y$-axis) using different noise levels ($X$-axis), with asymmetric (c) and asymmetric (d) noise models. As can be seen, ELP very significantly outperforms the other 2 scores when handling asymmetric noise.

## 6 SUMMARY AND DISCUSSION

We presented a new empirical observation, that the variability in the predictions of an ensemble of deep networks is much larger when labels are noisy, than it is when labels are clean. This observation is used as a basis for a new method for classification with noisy labels, addressing along the way the tasks of noise level estimation, noisy labels identification, and classifier construction. Our method is easy to implement, and can be readily incorporated into existing methods for deep learning with label noise, including semi-supervised methods, to improve the outcome of the methods.

Importantly, our method achieves this improvement without making additional assumptions, which are commonly made by alternative methods: (i) Noise level is expected to be unknown. (ii) There is no need for a clean validation set, which many other methods require, but which is very difficult to acquire when the training set is corrupted. (iii) Almost no additional hyperparameters are introduced.

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

APPENDIX

## A    OVERFIT AND INTER-MODEL CORRELATION

In this section we formally analyze the relation between two type of scores, which measure either overfit or inter-model agreement. *Overfit* is a condition that can occur during the training of deep neural networks. It is characterized by the co-occurring decrease of train error or loss, which is continuously minimized during the training of a deep model, and the increase of test error or loss, which is the ideal measure one would have liked to minimize and which determines the network's generalization error. An *agreement* score measures how similar the models are in their predictions.

We start by introducing the model and some notations in Section A.1. In Section A.2 we prove the main result (Prop. A.2): the occurrence of overfit at time s in all the models of the ensemble implies that the agreement between the models decreases.

### A.1    MODEL AND NOTATIONS

**Model.** We analyze the agreement between an ensemble of $Q$ models, computed by solving the linear regression problem with Gradient Descent (GD) and random initialization. In this problem, the learner estimates a linear function $f(\boldsymbol{x}) : \mathbb{R}^d \to \mathbb{R}$, where $\boldsymbol{x} \in \mathbb{R}^d$ denotes an input vector and $y \in \mathbb{R}$ the desired output. Given a training set of $M$ pairs $\{\boldsymbol{x}_m, y_m\}_{m=1}^M$, let $X \in \mathbb{R}^{d \times M}$ denote the training input - a matrix whose $m^{\text{th}}$ column is $\boldsymbol{x}_m \in \mathbb{R}^d$, and let row vector $\boldsymbol{y} \in \mathbb{R}^M$ denote the output vector whose $m^{\text{th}}$ element is $y_m$. Let $N$ denote the size of the test set. When solving a linear regression problem, we seek a row vector $\hat{\boldsymbol{w}} \in \mathbb{R}^d$ that satisfies

$$\hat{\boldsymbol{w}} = \text{argmin}_{\boldsymbol{w}} L(\boldsymbol{w}), \qquad L(\boldsymbol{w}) = \frac{1}{2}\|\boldsymbol{w}X - \boldsymbol{y}\|_F^2 \tag{3}$$

To solve (3) with GD, we perform at each iterative step $s \geq 1$ the following computation:

$$\boldsymbol{w}^{s+1} = \boldsymbol{w}^s - \mu\Delta\boldsymbol{w}^s$$

$$\Delta\boldsymbol{w}^s = \left.\frac{\partial L(\mathbb{X})}{\partial \boldsymbol{w}}\right|_{\boldsymbol{w}=\boldsymbol{w}^s} = \boldsymbol{w}^s\Sigma_{XX} - \Sigma_{YX} \qquad \Sigma_{XX} = XX^\top, \ \Sigma_{YX} = \boldsymbol{y}X^\top \tag{4}$$

for some random initialization vector $\boldsymbol{w}_0 \in \mathbb{R}^d$ where usually $\mathbb{E}[\boldsymbol{w}_0] = 0$, and learning rate $\mu$. Henceforth we omit the index $s$ when self evident from context.

As a final remark, when we use the notation $\|A\|$ below for some matrix $A$, differently from $\|A\|_F$, it denotes the operator norm of the symmetric matrix $A$, namely, its largest singular value.

**Additional notations**

- Index $i \in [Q]$ denotes a network instance, and $t$ denotes the test data. For simplicity and with some risk of notation abuse, let $Q$ and $Q'$ also denote sets of indices, either training or test. Specifically, $Q = [1, \ldots, Q]$ and $Q' = [1, \ldots, Q, t]$.
- We use function notation, where $\{X(i), y(i)\}$ is the training set of network $i$ and $\{X(t), y(t)\}$ is the test set. Thus

$$\Sigma_{XX}(j) = X(j)X(j)^\top, \quad \Sigma_{YX}(j) = \boldsymbol{y}(j)X(j)^\top \qquad j \in Q'$$

- Similarly, $\boldsymbol{w}(i) \in \mathbb{R}^d$ is the model learned by network $i$, and $\Delta\boldsymbol{w}(i)$ is the gradient step of $\boldsymbol{w}(i)$, where

$$\Delta\boldsymbol{w}(i) = \boldsymbol{w}(i)\Sigma_{XX}(i) - \Sigma_{YX}(i) \qquad i \in Q$$

- $\boldsymbol{e}(i, j)$ denotes a function, which maps indices $i \in Q, j \in Q'$ to the cross error of model $i$ on data $j$ - the classification error vector when using model $\boldsymbol{w}(i)$ to estimate $y(j)$. Let $M' = M$ if $j \in Q$ is a training index, and $M' = N$ if $j \in \{t\}$. Then we can write

$$\boldsymbol{e}(i, j) : Q \times Q' \to \mathbb{R}^{M'} \qquad \boldsymbol{e}(i, j) = \boldsymbol{w}(i)X(j) - \boldsymbol{y}(j)$$
$$\implies \qquad \Delta\boldsymbol{w}(i) = \boldsymbol{e}(i, i)X(i)^\top$$

  Note that in this notation, $\boldsymbol{e}(i, t)$ is the classification error vector when using model $i$, which is trained on data $X(i)$, to estimate the desired outcome on the test data - $y(t)$. $\|\boldsymbol{e}(i, t)\|_F$ is the test error, estimate of the generalization error, of classifier $i$.

- Let $\Delta(i,j)$ denote the cross gradient:

$$\Delta(i,j) = e(i,j)X(j)^\top = w(i)\Sigma_{XX}(j) - \Sigma_{YX}(j) \quad\Longrightarrow\quad \Delta w(i) = \Delta(i,i) \quad (5)$$

After each GD step, the model and the error are updated as follows:

$$\tilde{w}(i) = w(i) - \mu\Delta w(i)$$
$$\tilde{e}(i,j) = \tilde{w}(i)X(j) - y(j) = e(i,j) - \mu\Delta(i,i)X(j)$$

We note that at step $s$ and $\forall i,j \in Q$, $\tilde{w}(i)$ is a random vector in $\mathbb{R}^d$, and $\tilde{e}(i,j)$ is a random vector in $\mathbb{R}^M$. If $j \in \{t\}$, then $\tilde{e}(i,j) = \tilde{e}(i,t)$ is a random vector in $\mathbb{R}^N$.

**Test error random variable.** Using the above notations, $\{e(i,t)\}_{i=1}^{Q}$ is a set of $Q$ test errors vectors in $\mathbb{R}^N$, where the $n^{\text{th}}$ component of the $i^{\text{th}}$ vector $e(i,t)_n$ captures the test error of model $i$ on test example $n$. In effect, it is a sample of size $Q$ from the random variable $e(*,t)_n$. This random variable captures the error over test point $n$ of a model computed from a random sample of size $M$. The empirical variance of this random variable will be used to estimate the agreement between the models.

**Overfit.** Overfit occurs at step $s$ if

$$\|\tilde{e}(i,t)\|_F^2 > \|e(i,t)\|_F^2 \quad (6)$$

**Measuring inter-model agreement.** In classification problems, bi-modality of the ELP score captures the agreement between a set of classifiers, all trained on the same training matrix $X(i) = X$. Since here we are analyzing a regression problem, we need a comparable score to measure agreement between the predictions of $Q$ linear functions. This measure is chosen to be the variance of the test error among models. Accordingly, we will measure *disagreement* by the empirical variance of the test error random variable $\tilde{e}(*,t)_n$, average over all test examples $n \in [N]$.

More specifically, consider an ensemble of linear models $\{w(i)\}_{i=1}^{Q}$ trained on set $\mathbb{X}$ to minimize (3) with $s$ gradient steps, where $i$ denotes the index of a network instance and $Q$ the number of network instances. Using the test error vectors of these models $e(i,t)$, we compute the empirical variance of each element $\mathrm{var}[e(*,t)_n]$, and sum over the test examples $n \in [N]$:

$$\sum_{n=1}^{N} \sigma^2[e(*,t)_n] = \sum_{n=1}^{N} \frac{1}{2Q^2} \sum_{i=1}^{Q} \sum_{j=1}^{Q} |e(i,t)_n - e(j,t)_n|^2 = \frac{1}{2Q^2} \sum_{i=1}^{Q} \sum_{j=1}^{Q} \|e(i,t) - e(j,t)\|_F^2$$

**Definition 1** (Inter-model DisAgreement.). *The disagreement among a set of $Q$ linear models $\{w(i)\}_{i=1}^{Q}$ at step $s$ is defined as follows*

$$DisAg(s) = \frac{1}{2Q^2} \sum_{i=1}^{Q} \sum_{j=1}^{Q} \|e(i,t) - e(j,t)\|_F^2 \quad (7)$$

## A.2 OVERFIT AND INTER-NETWORK AGREEMENT

We first prove Lemma 1, which has the following intuitive interpretation: overfit occurs in model $i$ iff the gradient step of model $i$ (denoted $\Delta w(i)$), which is computed using the training set, is negatively correlated with the 'correct' gradient step - the one we would have obtained had we known the test set (this unattainable vector is denoted $\Delta(i,t)$).

**Lemma 1.** *Assume that the learning rate $\mu$ is small enough so that we can neglect terms that are $O(\mu^2)$. Then in each gradient descent step $s$, overfit occurs iff the gradient step $\Delta w(i)$ of network $i$ is negatively correlated with the cross gradient $\Delta(i,t)$.*

*Proof.* Starting from (6)

$$
\begin{aligned}
\text{(overfit)} &\iff \|\tilde{e}(i,t)\|_F^2 > \|e(i,t)\|_F^2 \\
&\iff \|\tilde{e}(i,t)\|_F^2 - \|e(i,t)\|_F^2 = \|e(i,t) - \mu\Delta(i,i)X(t)\|_F^2 - \|e(i,t)\|_F^2 > 0 \\
&\iff -2\mu\Delta(i,i)X(t)e(i,t)^\top + O(\mu^2) > 0 \\
&\iff \Delta(i,i) \cdot \Delta(i,t) < 0 \\
&\iff \Delta w(i) \cdot \Delta(i,t) < 0
\end{aligned}
\quad (8)
$$

$\square$

Lemma 2 claims that if the magnitude of the gradient step $\mu$ is small enough, then the operator norm of matrix $I - \mu\Sigma_{XX}$ is smaller than 1. The implication is that a geometric sum of this matrix converges, a technical result which will be used later.

**Lemma 2.** *For any invertible covariance matrix $\Sigma_{XX}$ there exists $\hat{\mu} > 0$, such that $\mu < \hat{\mu} \implies \|I - \mu\Sigma_{XX}\| < 1$.*

*Proof.* Since $\Sigma_{XX}$ is positive-definite, we can write $\Sigma_{XX} = USU^\top$ for orthogonal matrix $U$ and the diagonal matrix of singular values $S = diag\{s_i\}$. It follows that $I - \mu\Sigma_{XX} = Udiag\{1 - \mu s_i\}U^\top$, a matrix whose largest singular value is $1 - \mu s_d$. Since by assumption $s_d > 0$, the lemma follows. $\square$

Our last Lemma 3 claims that eventually, after sufficiently many gradient steps, the expected value of the solution is exactly the closed-form solution of the vetor that minimizes the loss.

**Lemma 3.** *Assume that $\|I - \mu\Sigma_{XX}\| < 1$ and $\Sigma_{XX}$ is invertible. If the number of gradient steps $s$ is large enough so that $\|I - \mu\Sigma_{XX}\|^s$ can be neglected, then*

$$\mathbb{E}[\boldsymbol{w}^s] \approx \Sigma_{YX}\Sigma_{XX}^{-1} \tag{9}$$

*Proof.* Starting from (4), we can show that

$$\boldsymbol{w}^s = \boldsymbol{w}^0(I - \mu\Sigma_{XX})^{s-1} + \mu\Sigma_{YX}\sum_{k=1}^{s-1}(I - \mu\Sigma_{XX})^{k-1}$$

Since $\mathbb{E}(\boldsymbol{w}^0) = 0$

$$\mathbb{E}(\boldsymbol{w}^s) = \mathbb{E}(\boldsymbol{w}^0)(I - \mu\Sigma_{XX})^{s-1} + \mu\Sigma_{YX}\sum_{k=1}^{s-1}(I - \mu\Sigma_{XX})^{k-1} = \mu\Sigma_{YX}\sum_{k=1}^{s-1}(I - \mu\Sigma_{XX})^{k-1}$$

Given the lemma's assumptions, this expression can be evaluated and simplified:

$$\begin{aligned} \mathbb{E}(\boldsymbol{w}^s) &= \mu\Sigma_{YX}[I - (I - \mu\Sigma_{XX})]^{-1}[I - (I - \mu\Sigma_{XX})^{s-1}] \\ &= \Sigma_{YX}\Sigma_{XX}^{-1} - \Sigma_{YX}\Sigma_{XX}^{-1}(I - \mu\Sigma_{XX})^{s-1} \\ &\approx \Sigma_{YX}\Sigma_{XX}^{-1} \end{aligned} \tag{10}$$

$\square$

From (7) it follows that a decrease in inter-model agreement at step $s$, which is implied by increased test variance among models, is indicated by the following inequality:

$$\begin{aligned} \mathbb{C} &= DisAg(s) - DisAg(s-1) \\ &= \frac{1}{2Q^2}\sum_{i,j=1}^{Q}\|\tilde{\boldsymbol{e}}(i,t) - \tilde{\boldsymbol{e}}(j,t)\|_F^2 - \frac{1}{2Q^2}\sum_{i,j=1}^{Q}\|\boldsymbol{e}(i,t) - \boldsymbol{e}(j,t)\|_F^2 > 0 \end{aligned} \tag{11}$$

**Theorem.** *Assume that all models see the same training set, denoted as $X(i) = X \ \forall i \in [Q]$, and that the training data covariance matrix $\Sigma_{XX}$ is full rank.*

*We make the following asymptotic assumptions, which are loosely phrased but can be rigorously defined with additional notations:*

1. *The learning rate $\mu$ is small enough so that $\|I - \mu\Sigma_{XX}\| < 1$ (from Lemma 2), and additionally we can neglect terms that are $O(\mu^2)$.*
2. *The number of gradient steps $s$ is large enough so that $\|I - \mu\Sigma_{XX}\|^s$ can be neglected.*
3. *The number of models $Q$ is large enough so that using the law of large numbers, we get $\frac{1}{Q}\sum_{i=1}^{Q}\boldsymbol{w}(i) \approx \mathbb{E}[\boldsymbol{w}]$.*

*Finally, we assume that overfit occurs at time $s$ in all the models of the ensemble. In other words, at time $s$ the generalization error does not decrease in all the models.*

*When these assumptions hold, the agreement between the models decreases.*

*Proof.* (11) can be rearranged as follows

$$\mathbb{C} = \frac{1}{2Q^2} \sum_{i,j=1}^{Q} \|[\boldsymbol{e}(i,t) - \mu\Delta(i,i)X(t)] - [\boldsymbol{e}(j,t) - \mu\Delta(j,j)X(t)]\|_F^2 - \frac{1}{2Q^2} \sum_{i,j=1}^{Q} \|\boldsymbol{e}(i,t) - \boldsymbol{e}(j,t)\|_F^2$$

$$= \frac{1}{Q^2} \sum_{i,j=1}^{Q} -\mu[\boldsymbol{e}(i,t) - \boldsymbol{e}(j,t)] \cdot [\Delta(i,i)X(t) - \Delta(j,j)X(t)] + O(\mu^2)$$

$$= \frac{\mu}{Q^2} \sum_{i,j=1}^{Q} [\Delta(i,i) \cdot \Delta(j,t) + \Delta(j,j) \cdot \Delta(i,t)] - [\Delta(i,i) \cdot \Delta(i,t) + \Delta(j,j) \cdot \Delta(j,t)] + O(\mu^2)$$

where the last transition follows from $\boldsymbol{e}(i,t)X(t)^\top = \Delta(i,t)$. Using assumption 2

$$\mathbb{C} = \mu(\mathbb{C}' - \mathbb{C}'') + O(\mu^2) \approx \mu(\mathbb{C}' - \mathbb{C}'') \tag{12}$$

where

$$\mathbb{C}'' = \frac{1}{Q^2} \sum_{i,j=1}^{Q} [\Delta(i,i) \cdot \Delta(i,t) + \Delta(j,j) \cdot \Delta(j,t)] = \frac{2}{Q} \sum_{i=1}^{Q} \Delta(i,i) \cdot \Delta(i,t) \tag{13}$$

and

$$\mathbb{C}' = \frac{1}{Q^2} \sum_{i,j=1}^{Q} [\Delta(i,i) \cdot \Delta(j,t) + \Delta(j,j) \cdot \Delta(i,t)]$$

$$= \frac{1}{Q} \sum_{i=1}^{Q} \Delta(i,i) \cdot \frac{1}{Q} \sum_{j=1}^{Q} \Delta(j,t) + \frac{1}{Q} \sum_{j=1}^{Q} \Delta(j,j) \cdot \frac{1}{Q} \sum_{i=1}^{Q} \Delta(i,t) \tag{14}$$

$$= \frac{1}{Q} \sum_{i=1}^{Q} \Delta(i,i) \cdot \frac{2}{Q} \sum_{j=1}^{Q} \Delta(j,t)$$

Next, we prove that $\mathbb{C}'$ is approximately 0. We first deduce from assumptions 1 and 4 that

$$\frac{1}{Q} \sum_{i=1}^{Q} \Delta(i,i) = \frac{1}{Q} \sum_{i=1}^{Q} \boldsymbol{w}(i)\Sigma_{XX}(i) - \Sigma_{YX}(i) = \left(\frac{1}{Q} \sum_{i=1}^{Q} \boldsymbol{w}(i)\right) \Sigma_{XX} - \Sigma_{YX} \approx \mathbb{E}[\boldsymbol{w}]\Sigma_{XX} - \Sigma_{YX}$$

From assumption 3 and Lemma 3, we have that $\mathbb{E}[\boldsymbol{w}] \approx \Sigma_{YX}\Sigma_{XX}^{-1}$. Thus

$$\frac{1}{Q} \sum_{i=1}^{Q} \Delta(i,i) \approx \mathbb{E}[\boldsymbol{w}]\Sigma_{XX} - \Sigma_{YX} \approx \Sigma_{YX}\Sigma_{XX}^{-1}\Sigma_{XX} - \Sigma_{YX} = 0$$

From this derivation and (14) we may conclude that $\mathbb{C}' \approx 0$. Thus

$$\mathbb{C} \approx -\mu\mathbb{C}'' = -\mu\frac{2}{Q} \sum_{i=1}^{Q} \Delta(i,i) \cdot \Delta(i,t) \tag{15}$$

If overfit occurs at time $s$ in all the models of the ensemble, then $\mathbb{C} > 0$ from Lemma 1 and (15). From (11) we may conclude that the inter-model agreement decreases, which concludes the proof.

$\square$

## B   NOISY LABELS AND INTER-MODEL AGREEMENT

Here we show empirical evidence, that noisy labels are **not** being learned in the same order by an ensemble of networks. To see this, we measure the distance between the TPA distribution, computed separately for clean examples and for noisy examples, and the binomial distribution, which is the

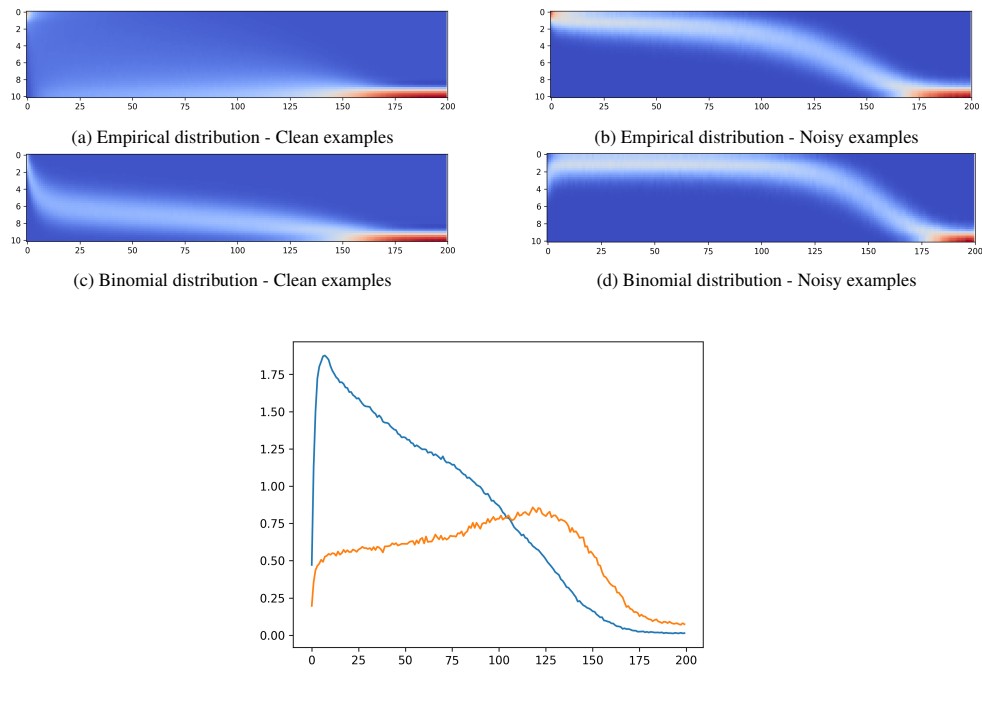

(a) Empirical distribution - Clean examples

(b) Empirical distribution - Noisy examples

(c) Binomial distribution - Clean examples

(d) Binomial distribution - Noisy examples

(e)

Figure 6: (a)-(d): The empirical distribution of the agreement values over epochs ($X$-axis: epochs, $Y$-axis: agreement, color code: blue for low and red for high). Clearly, the distribution of noisy examples resembles the binomial distribution with matched expected value, while the clean examples distribution is far from binomial. (e) Wasserstein distance between the binomial distribution and the empirical agreement distribution over epochs.

expected distribution of iid classifiers with the same overall accuracy. Specifically, we compute the Wasserstein distance between the agreement distribution at each epoch and the binomials BIN (k,$p_{clean}$) and BIN(k,$p_{noisy}$), where $p_{clean}$ is the average accuracy on the clean examples, and $p_{noisy}$ is the average accuracy on the noisy examples, see Fig. 6. We see that while the distribution of model agreement on clean examples is very far from the binomial distribution, the distribution of model agreement on noisy examples is much closer, which means that each DNN memorizes noisy examples at an approximately independent rate.

## C  COMPUTING THE ELP SCORE: PSEUDO-CODE

## D  ADDITIONAL RESULTS

### D.1  NOISE LEVEL ESTIMATION ON ADDITIONAL DATASETS

---

**Algorithm 2:** Computing the ELP score

---

**Input:** Training dataset $\mathbb{X}$ with $N$ examples, potentially noisy, network architecture $\mathcal{A}$, batch size $b$, learning rate $\eta$, number of networks K, number of epochs E
**Output:** Array, containing the ELP score for each data point
compute agreement during training;
**initialization** $\theta_1^0...\theta_K^0$ different initialization of $\mathcal{A}$;
Initialize *agreement_arr*[E,N]$\leftarrow 0$;
**for** $e = 0; E$ **do**
    **for** $k = 0; K$ **do**
        sample $indices_b$ with size $b$ from [1...N] uniformly;
        $x_b, y_b \leftarrow \mathcal{X}[indices_b], \mathcal{Y}[indices_b]$;           `/* Gets a mini batch */`
        compute $p_b$ on $x_b$ using $\theta_k^e$;
        compute loss $l_b$ with respect to $p_b$ and $y_b$;
        $\theta_k^{e+1} \leftarrow SGD(\theta_k^e; l_b)$;
        $agreement\_arr[e, indice_b]$ **+=** (**argmax**$(p_b) == y_b$);   `/* Store whether the`
        `network k predicted correctly on the examples at epoch e`
        `*/`
    **end**
**end**
$agreement\_arr \leftarrow agreement\_arr/(E \cdot K)$;           `/* normaliztion */`
$ELP\_arr \leftarrow mean\_over\_epochs(agreement\_arr)$;     `/* mean over $X$-axis */`
**return** $ELP\_arr$

---

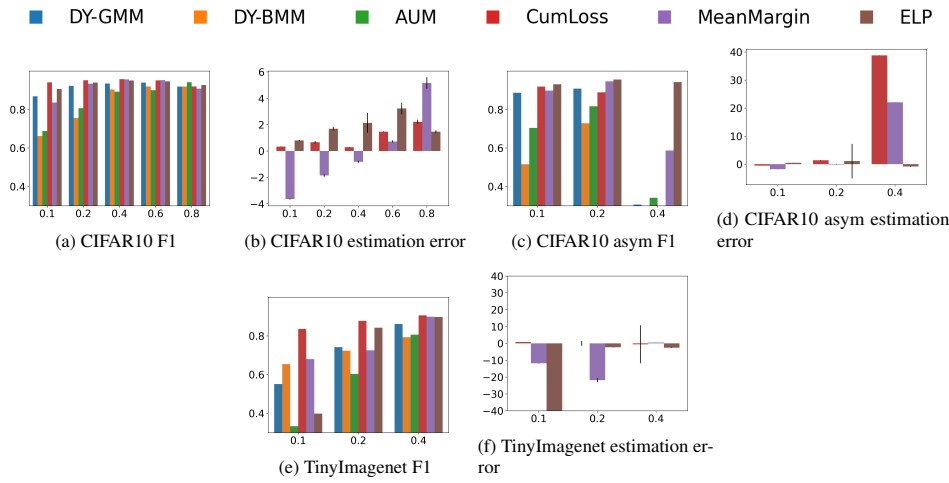

Figure 7: Additional results on CIFAR10 and Tiny Imagenet.

## D.2 PRECISION AND RECALL RESULTS

See fig. 8

## E ABLATION STUDY

Table 3 summarize experiments relating to architecture, scheduler, and augmentation usage. Table 4 provides additional support that our method can be used with only a few networks, as the test accuracy after retraining does not improve when using more than 2-3 networks. In table 5 we compared our method to two compatative methods from recent years that use semi-supervised learning or similar ideas (Li et al., 2020; Liu et al., 2020). These methods have a different goal than ours (constructing a robust classifier, and not noisy labels identification), which is why we did not compare to them in the main text. However, we chose to include this comparison here, as it shows that semi-supervised

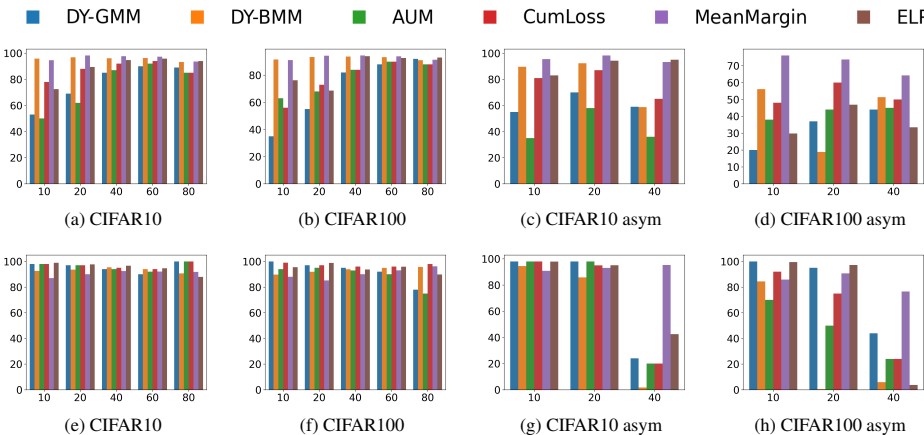

Figure 8: Noisy label identification. Top: precision; bottom: recall.

methods are not always stable and robust as a solution for label noise, and sometimes its best to identify the noisy labels and retrain on the filtered dataset.

Table 3: F1 score for Cifar100 with 2 levels of symmetric noise. Different ablation conditions are marked in columns: *ResNet34* indicates a change of architecture, *no Aug* indicates that image augmentations are not used, and *lr 0.01* indicates that no scheduler or learning rate drop are used during training.

| Noise level | No change | ResNet34 | No Aug | Constant lr 0.01 | Lr 0.01 + no Aug |
|---|---|---|---|---|---|
| 20% | $0.903 \pm 0.01$ | $0.839 \pm 0.0$ | $0.832 \pm 0.0$ | $0.859 \pm 0.01$ | $0.869 \pm 0.01$ |
| 40% | $0.918 \pm 0.01$ | $0.887 \pm 0.0$ | $0.887 \pm 0.01$ | $0.877 \pm 0.0$ | $0.888 \pm 0.01$ |

Table 4: test accuracy after retraining on filtered dataset when using DisagreeNet with different numbers of models.

| Dataset | Noise | size of ensemble | | |
|---|---|---|---|---|
| Method | | 2 | 3 | 10 |
| | 20% | $93.8 \pm 0.1$ | $93.7 \pm 0.1$ | $93.1 \pm 0.2$ |
| Cifar10 sym | 40% | $91.5 \pm 0.1$ | $91.1 \pm 0.1$ | $91.1 \pm 0.1$ |
| | 60% | $86.4 \pm 0.2$ | $86.1 \pm 0.3$ | $83.9 \pm 0.08$ |
| | 20% | $76.3 \pm 0.2$ | $76.2 \pm 0.2$ | $77.3 \pm 0.2$ |
| Cifar100 sym | 40% | $72.0 \pm 0.5$ | $71.8 \pm 0.4$ | $71.8 \pm 0.3$ |
| | 60% | $64.1 \pm 0.05$ | $64.3 \pm 0.2$ | $64.7 \pm 0.3$ |
| | 20% | $93.1 \pm 0.1$ | $93.6 \pm 0.1$ | $94.4 \pm 0.1$ |
| Cifar10 asym | 40% | $90.9 \pm 0.1$ | $91.25 \pm 0.2$ | $91.9 \pm 0.0$ |
| | 20% | $74.8 \pm 0.1$ | $74.86 \pm 0.1$ | $73.9 \pm 0.5$ |
| Cifar100 asym | 40% | $59.9 \pm 0.1$ | $59.7 \pm 0.2$ | $61.3 \pm 0.2$ |

**Alternative scores** We evaluate the two alternative scores defined in Section 2.3: CumLoss and MeanMargin, in which case Step 2 of *DisagreeNet* is executed using one of them instead of the ELP score. Fig. 9 shows the Probability Distribution Function (PDF) of the three scores, revealing that ELP is more consistency bimodal (especially in the difficult asymmetric case), with modes (peaks) that appear more separable. This benefit translates to superior performance in the noise filtration task (Figs. 5b,5d). In addition, we evaluated the importance of using solely the last epoch instead of the entire history in Fig. 10, which shows that using ELP is superior to using the TPA for filtration, even if one uses the best epoch for TPA filtration.

We believe that this empirical observation, of increased mode separation, is due to significant difference in the pace of change in agreement values during training between clean and noisy data, in contrast with the pace of change in smoother measures of confidence like *Margin* and *Loss* (see App. G). Note that with the easier symmetric noise, we do not see this difference, and indeed the other scores exhibit two nicely separated modes, sometimes achieving even better results in noise

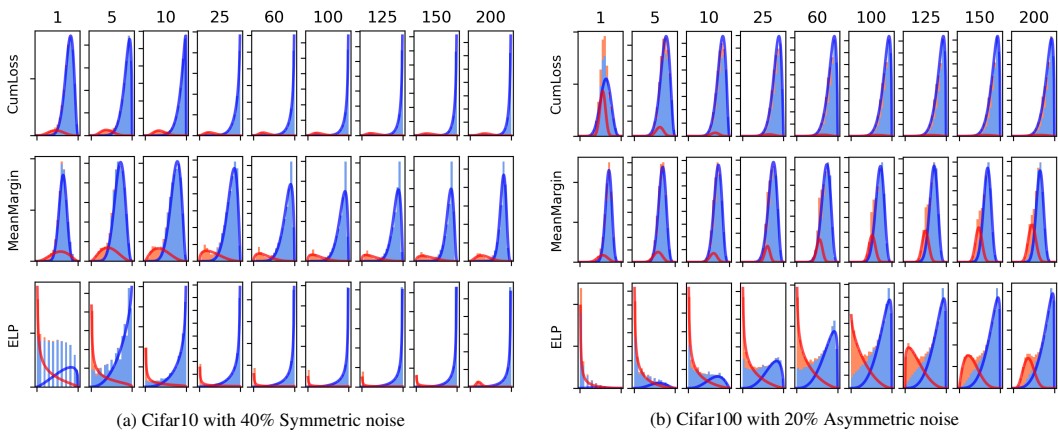

(a) Cifar10 with 40% Symmetric noise      (b) Cifar100 with 20% Asymmetric noise

Figure 9: Distribution of the CumLoss, MeanMargin and ELP scores during training. ELP remains bimodal even for hard noise models, where the other scores become unimodal.

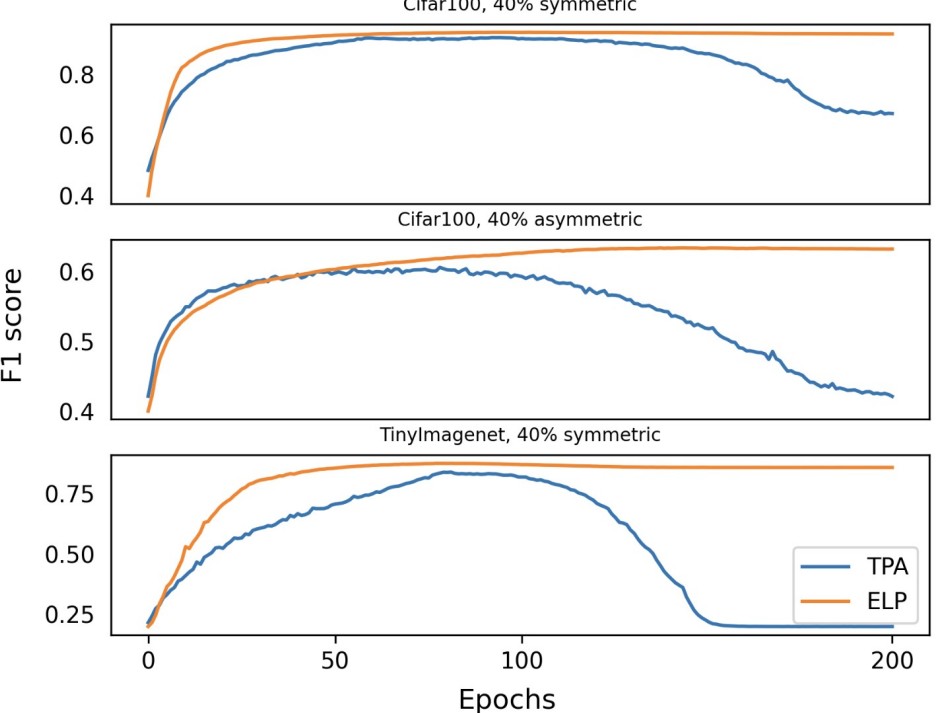

Figure 10: F1 score over the epochs, when using the entire history (ELP) or just the last epoch (TPA), evaluated with prior knowledge

Table 5: Test accuracy (%), average and standard error, in the best epoch of retraining after filtration. Results of ELR and DivideMix were computed by us, using the official implementation for the methods, using same hyperparameters as ours.

| Method/**Dataset** | **Tiny Imagenet sym** | |
|---|---|---|
| Noise level | 20% | 40% |
| *DivideMix (DenseNet)* | $39.88 \pm 0.3$ | $37.02 \pm 0.3$ |
| *ELR (DenseNet)* | $63.8 \pm 0.0$ | $57.4 \pm 0.2$ |
| *DisagreeNet*+SL | $\mathbf{64.5 \pm 0.1}$ | $\mathbf{58.5 \pm 0.2}$ |

Table 6: Final accuracy results when changing the backbone architecture.

| Method/**Dataset** | **CIFAR-10 sym** | | | **CIFAR-100 sym** | | |
|---|---|---|---|---|---|---|
| Noise level | 20% | 40% | 60% | 20% | 40% | 60% |
| *DisagreeNet+SL R18* | $93.3 \pm 0.1$ | $91.1 \pm 0.6$ | $87.6 \pm 0.1$ | $75.1 \pm 0.1$ | $71.5 \pm 0.1$ | $63.1 \pm 0.4$ |
| *DisagreeNet+SL R50* | $93.4 \pm 0.1$ | $91.0 \pm 0.2$ | $87.0 \pm 0.1$ | $75.7 \pm 0.3$ | $70.2 \pm 1.2$ | $61.0 \pm 0.4$ |

filtration than ELP (Fig. 7 in App. D.1). However, when comparing the test accuracy after retraining (see App. D), we observe that ELP still achieves superior results.

Additionally, in table 7 we evaluate the confusion score in (Simsek et al., 2022), which is the mean (over the epochs) entropy of the mean (over multiple networks) logit of each class (used in the original paper for our of distribution study). We see that this score is also much inferior to ours in the binary classification task of noisy labels identification.

Table 7: AUC (area under the curve) score for noisy labels identification using our ELP score and the confusion score from (Simsek et al., 2022)

| Method/**Dataset** | **CIFAR-10 sym** | | **CIFAR-100 sym** | |
|---|---|---|---|---|
| Noise level | 20% | 40% | 20% | 40% |
| *Confusion* | 0.809 | 0.819 | 0.820 | 0.824 |
| *ELP*+SL | **0.997** | **0.994** | **0.991** | **0.986** |

# F  METHODOLOGY AND TECHNICAL DETAILS

**Datasets** We evaluated our method on a few standard image classification datasets, including Cifar10 and Cifar100 (Krizhevsky et al., 2009) and Tiny imagenet (Le and Yang, 2015). Cifar10/100 consist of 60k $32 \times 32$ color images of 10 and 100 classes respectively. Tiny ImageNet consists of 100,000 images from 200 classes of ImageNet (Deng et al., 2009), downsampled to size $64 \times 64$. Animal10N dataset contains 5 pairs of confusing animals with a total of 55,000 64x64 images. Clothing1M (Xiao et al., 2015) contains 1M clothing images in 14 classes. These datasets were used in earlier work to evaluate the success of noise estimation (Pleiss et al., 2020; Arazo et al., 2019; Li et al., 2020; Liu et al., 2020).

**Baseline methods for comparison** We evaluate our method in the context of two approaches designed to deal with label noise: methods that focus on improving the supervised learning by identifying noisy labels and removing/reducing their influence on the training, and methods that use iterative methods and utilize semi-supervised algorithms in order to learn with noisy labels. **First approach:** ◇ **DY-BMM** and **DY-GMM** (Arazo et al., 2019) estimate mixture models on the loss to separate noisy and clean examples. ◇ **INCV** (Chen et al., 2019) iteratively filter out noisy examples by using cross-validation. ◇ **AUM** (Pleiss et al., 2020) inserts corrupted examples to determine a filtration threshold, using the mean margin as a score. ◇ **Bootstrap** (Reed et al., 2014) interpolates between the net predictions and the given label. ◇ **D2L** (Ma et al., 2018) follows *Bootstrap*, and uses the examples dimensional attributes for the interpolation. ◇ **Co-teaching** (Han et al., 2018) use two networks to filter clean data for the other net training. ◇ **O2U** (Huang et al., 2019b) varies the learning rate to identify the noisy samples, based on a loss-based metric. ◇ **MentorNet** (Jiang et al., 2018) trains a mentor network, whose outputs are used as a curriculum to the student network. ◇ **LEC** (Lee and Chung, 2019) trains multiple networks, and uses the intersection of their small loss examples (using a given noise rate as a threshold) to construct a filtered dataset for the next epoch.

**Second approach:** ⋄ **SELF** (Nguyen et al., 2019) iteratively uses an exponential moving average of a net prediction over the epochs, compared to the ground truth labels, to filter noisy labels and retrain . ⋄ **Meta learning** (Li et al., 2019) uses a gradient based technique to update the networks weights with noise tolerance. ⋄ **DivideMix** (Li et al., 2020) uses 2 networks to flag examples as noisy and clean with two component mixture, after which the SSL technique MixMatch (Berthelot et al., 2019) is used. ⋄ **ELR** (Liu et al., 2020) identifies early learned example, and uses them to regulate the learning process. ⋄ **C2D** (Zheltonozhskii et al., 2022) uses the same algorithm as ELR and Dividemix, and uses a pretrain net with unsupervised loss.

**Technical Details** Unless stated otherwise, we used an SGD optimizer with 0.9 momentum and a learning rate of 0.01, weight decay of 5e-4, and batch size of 32. We used a Cosine-annealing scheduler in all of our experiments and used standard augmentation (horizontal flips, random crops) during training. We inspected the effect of different hyperparameters in the ablation study. All of our experiments were conducted on the internal cluster of the Hebrew University, on GPU type AmpereA10.

## G    COMPARING AGREEMENT TO CONFIDENCE IN NOISE FILTRATION

While the learning time of an example has been shown to be effective for noise filtration, it fails to separate noisy and clean data that are learned more or less at the same time. To tackle this problem, one needs additional information, beyond the learning time of a single network. When using an ensemble, we can use the TPA score, or else the average probability assigned to the ground truth label (denoted the "correct" logit) by the networks. The latter score conveys the model's confidence in the ground truth label, and is used by our two alternative scores - CumLoss and MeanMargin.

Going beyond learning time, we propose to look at "how quickly" the agreement value rises from 0 to 1, denoted as the "slope" of the agreement. Since our empirical results indicate that the learning time of noisy data is much more varied, we expect a slower rise in agreement over noisy data as compared to clean data. In our experiments, ELP achieved superior results in noise filtration. We hypothesize that the difference in slope between clean and noisy data may underlie the superiority of ELP in noise filtration.

To check this hypothesis, we compare between two scores computed at each data example: ELP and Logits Mean (denoted LM for simplicity). LM is defined as follows:

$$LM(x) = \frac{\sum_{i=1}^{k} \sum_{j=1}^{T} [p_{i,j}(x)]_y}{kT}$$

where $k$ is the number of networks, $T$ is the number of epochs during training, $(x, y)$ is a data example and its assigned label, and $[p_{i,j}(x)]_y$ is the probability assigned by network $i$ in epoch $j$ to $y$ (the ground truth label).

In order to compare between the pace of increase (slope) of ELP and LM, we conduct the following analysis: We select the two groups of clean and noisy data that are learned (roughly) at the same time by some net in the ensemble, and then compute the average agreement and "correct" logit functions as a function of epoch, separately for clean and noisy data. We then compute the difference per epoch between the noisy and clean average agreement, which we denote as $\Delta Agreement$ and $\Delta logit$. Note that $\Delta Agreement$ and $\Delta logit$ encode the difference in the slope between noisy and clean data, since they begin to rise at (roughly) the same time. Finally, we plot in Fig. 11 the difference between $\Delta Agreement$ and $\Delta logit$, recalling that larger $\Delta$ indicates stronger separation between the clean and noisy data.

Indeed, our analysis shows that with asymmetric noise, the difference between the agreement slope on clean and noisy data of the ELP score is consistently larger than the agreement slope difference between the average logits on clean and noisy data. This, we believe, is the key reason as to why ELP outperforms LM in noise filtration. Note that this effect is much less pronounced when using the easier symmetric noise, and indeed, our empirical results show that ELP does not outperform LM significantly in this case.

To conclude, we believe that the signal encoded by the agreement values is stronger than the signal encoded in measures of confidence in the networks' prediction when true labels are concerned, which

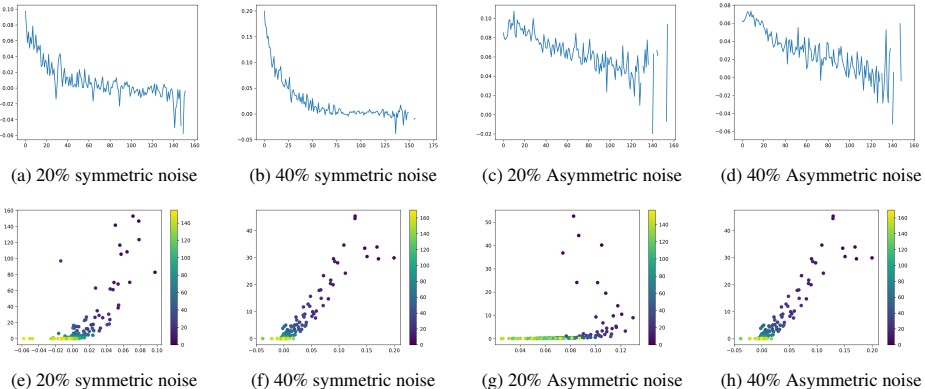

Figure 11: Top: $X$-axis is the learning time of the chosen clean and noisy data; $Y$-axis is the difference between $\Delta Agreement$ and $\Delta logit$. We see that for most of the training, the difference is positive, implying that ELP provides stronger separation between these groups. Bottom: $X$-axis is the difference between $\Delta Agreement$ and $\Delta logit$; $Y$-axis is the ratio between the amount of clean and noisy data. The color represents the learning time of the groups. These graphs show that while at the end of the training the difference between $\Delta Agreement$ and $\Delta logit$ is negative, implying that LM would be better at separating these groups, these are in fact very small sets of data, as most of the data is learned by some network at an earlier stage of the training

explains its capability to classify correctly even some hard-clean examples and easy-noisy examples as clean and noise (respectively). This, we believe, is a result of the polarization effect caused by the binary indicators inside TPA, which disregard misleading positive probabilities assigned to noisy labels even before they are learned by the networks.

## H  COMPARING TO METHODS WITH DIFFERENT ASSUMPTIONS

Here we compare DisagreeNet to methods that assume known noise level - O2U (Huang et al., 2019b) and LEC (Lee and Chung, 2019), using a 9-layered CNN for the training (with standard hyper parameters as detailed in App. F). Since the noise level is assumed known, we replace the estimation provided by DisagreeNet with the actual noise level. The results are summarized in Table. 8. We also compare DisagreeNet to other methods that use prior knowledge, where DisagreeNet does not use prior knowledge. The results are summarized in Table. 9

| Dataset | Noise level | Method | | |
|---|---|---|---|---|
| Dataset - noise type | Noise level | O2U | LEC | DisagreeNet |
| Cifar10 sym | 10% | 87.64% | - | **92.57%** |
| | 20% | 85.24% | 88.31% | **91.44%** |
| | 40% | 79.64% | - | **88.48%** |
| | 60% | - | 80.52%. | **81.81%** |
| Cifar100 sym | 10% | 62.32% | - | **69.86%** |
| | 20% | 60.53% | 59.98% | **67.99%** |
| | 40% | 52.47% | - | **62.89%** |
| | 60% | - | 46.63% | **53.76%** |
| Cifar10 asym | 10% | 88.22% | - | **91.96%** |
| | 20% | - | 89.41%. | **90.94%** |
| | 40% | - | 86.50% | **86.93%** |
| Cifar100 asym | 10% | 64.50% | - | **69.83%** |
| | 20% | - | 58.86%. | **67.99%** |
| | 40% | - | 47.82%. | **62.89%** |

Table 8: Test accuracy (%) comparison with methods that utilize prior knowledge with 9-layered CNN

| Method/**Dataset** | **CIFAR-10 sym** | | | **CIFAR-100 sym** | | |
|---|---|---|---|---|---|---|
| Noise level | 20% | 40% | 60% | 20% | 40% | 60% |
| Co-teaching | $88.8 \pm 0.1$ | $86.5 \pm 0.1$ | $80.7 \pm 0.1$ | $64.1 \pm 0.1$ | $60.2 \pm 0.2$ | $48.0 \pm 0.3$ |
| LEC | 88.3 | - | 80.5 | 60 | - | 46.63 |
| O2U | 92.5 | 90.3 | - | 74.1 | 69.2 | - |
| *DisagreeNet*+SL (no prior knowledge) | $\mathbf{93.1 \pm 0.2}$ | $\mathbf{91.1 \pm 0.1}$ | $\mathbf{83.9 \pm 0.08}$ | $\mathbf{77.3 \pm 0.2}$ | $\mathbf{71.8 \pm 0.3}$ | $\mathbf{64.7 \pm 0.3}$ |

| Method/**Dataset** | **CIFAR-10 asym** | | **CIFAR-100 asym** | | **Tiny Imagenet sym** | |
|---|---|---|---|---|---|---|
| Noise level | 20% | 40% | 20% | 40% | 20% | 40% |
| *LEC* | 89.4 | 86.5 | 58.9 | 47.8 | - | - |
| *DisagreeNet*+SL (no prior knowledge) | $\mathbf{94.4 \pm 0.1}$ | $\mathbf{91.9 \pm 0.0}$ | $\mathbf{73.9 \pm 0.5}$ | $\mathbf{61.3 \pm 0.2}$ | $\mathbf{62.5 \pm 0.2}$ | $\mathbf{55.7 \pm 0.4}$ |

| Method/**Dataset** | **animal10N, 8% noise** | |
|---|---|---|
| Noise level | noise est | test accuracy |
| *Co-teaching* | - | $82.5 \pm 0.1$ |
| *SELFIE* | - | $83 \pm 0.1$ |
| *DisagreeNet*+SL (no prior knowledge) | 7.8 | $\mathbf{85.1 \pm 0.1}$ |

Table 9: Test accuracy (%) comparison with methods that utilize prior knowledge of the real noise level.

