# OpenReview forum: "The Dynamic of Consensus in Deep Networks and the Identification of Noisy Labels"
_ICLR.cc/2023/Conference — Submitted to ICLR 2023_

### Official Review · Reviewer_thYY · 2022-10-20

**Confidence:** 3
**Correctness:** 2
**Technical Novelty And Significance:** 3
**Empirical Novelty And Significance:** 3
**Recommendation:** 5

**Clarity, Quality, Novelty And Reproducibility:**

Clarity: Good

Quality: Boardline, the major limition is in the empirical

Novelty: Boardine, see the discussion in the major weakness

Reproducibility: I haven't check the code

**Strength And Weaknesses:**

Strength:
* The algorithm is inspired by interesting observations.
* The paper is well written and easy to follow.

Major Weaknesses:
* Though existing works haven't explicity discuss / show how the emsemble models perform on the noisy data (disagree/agree with each other), the idea itself is not surprising. Because it is well known that the models are less confident on the noisy data samples, so we can expect the models that start from different initializations will disagree with each other. The famous co-teaching series of works have already built lots algorithms on the fact that models are less certain on noisy data.
* The empirical results are weak, e.g., it only has 70.8 on Clothing1M. Much inferior to existing works, e.g., VolMinNet 72.42 [1].
[1] Provably End-to-end Label-noise Learning without Anchor Points

Minor Issue:
* I suggest to use "Theorem(informal)" in Page 4.

Question:
* On the theretical part: intuitively, why the agreement would decrease when overfitting occurs? Since the models overfit the data, they will predict the noisy label perfectly because all models remember the data. For example, given any $(x_i, y_i)$ in a dataset,for any function $f$ that overfits the dataset, it means $f(x_i) = y_i$.

**Summary Of The Paper:**

This paper introduce an interesting idea to distinguish between noisy and clean samples. Specifically, they start from the observation that an emsemble of models disagree on the noisy samples during training. Based on this, the author further propose to select the clean samples according to the consistency between models. The proposed method is justfied by a theoretical analysis and empirical experiments.

**Summary Of The Review:**

In summary of the discussion above,  I suggest the authors to enhance the novlty of the proposed algorithm and empirical perfomance.  Personally, I tend to give a boardline reject recommendation.

---

### Official Review · Reviewer_skxN · 2022-10-24

**Confidence:** 5
**Correctness:** 2
**Technical Novelty And Significance:** 2
**Empirical Novelty And Significance:** 2
**Recommendation:** 3

**Clarity, Quality, Novelty And Reproducibility:**

Clarity:
The writing and presentation of the paper need improvement. E.g., for figure 2, which demonstrates the major observation, I think it would be much more intuitive to draw the test-accuracy/loss v.s. epoch curve alone with the BI v.s. epoch curve rather than showing the relationship in the side plot figure 2 (b). Moreover, it is not clear to me the differences between the top and the bottom plot in figure 2 (b). There are also many other places in the paper that require careful modifications (see some in minor comments).

Quality:
The overall quality is OK. For the experiment, in particular, since the proposed method only identifies the noisy labels, it is hard to compare the proposed results to other noisy label learning methods that fully utilize the entire training dataset (e.g., the results reported in the MentorMix paper, the network used may be another factor). To make the results more convincing, I think it is necessary to show the results in settings more comparable to previous work, for example, using the same network and tasks.

Moreover, since the proposed method uses significantly more computation for noisy-label identification, it may be unfair for other baselines. The authors may need to justify the computation costs further or compare baseline methods with larger models.

Novelty:
As discussed above int he strengths and weaknesses part, the novelty of the paper is limited.

Reproducibility:
There are many details given and I think the reproducibility is trustworthy.

Minor Comments:
Section 1: "Relation to prior art", maybe change it to a subsection?
Section 3: "’overfit’", the quotes are not used correctly.
Section 5: the proposed method is often named as "ELP" in the figures but referred to as disagree nets in the text.
Section 5: figure 4, what is the y-axis?
Section 5: figure 5, the legend fonts are too large.

**Strength And Weaknesses:**

Strengths:
1. The paper shows an interesting observation and correlates the identification of noisy labels to the disagreements of an ensemble of models.

2. The paper proposes a very simple method to identify noisy-labeled samples based on observations about disagreements among an ensemble of models.

Weaknesses:
1. The novelty of the paper is limited. Many previous papers share similar spirits that disagreements among multiple models are utilized to identify noisy labels. For example, a line of research including co-teaching, co-teaching+ and etc uses two models to help find noisy labels. Also, SELF uses the self-ensembling of one model at various training epochs to identify noises.

2. The proposed method has significantly increased computational costs as it requires the training of an ensemble of models. Based on results in Section 5.4, it seems that as many as 7 models are required to get the best performance.

3. The proposed method only considers the problem of noisy label identification for noisy label learning. It relies on other methods to utilize the noisy-labeled samples. Moreover, it is not clear how to utilize the method adaptively, i.e., identify certain noisy labels, train the model using the correct labels, and re-identify more labels with the better-trained models.

**Summary Of The Paper:**

The paper studies the problem of noisy label learning and proposes a method that computes a score based on the agreements over an ensemble of models to identify the samples with the correct given labels. The paper conducts experiments and shows that the disagreements among the ensemble of models strongly correlate with the overfitting behavior, indicating that the labels could be noisy. The paper utilizes such an observation to identify samples with clean labels and use other noisy-label learning or semi-supervised learning method to do the training.

**Summary Of The Review:**

Given the limitations in the novelty, the problems in the experiments, and the overall presentation of the paper, I don't think the paper is ready to be published for now.

---

### Official Review · Reviewer_ft5z · 2022-10-24

**Confidence:** 4
**Clarity, Quality, Novelty And Reproducibility:** 1. The paper is well written, with ve…
**Correctness:** 2
**Technical Novelty And Significance:** 2
**Empirical Novelty And Significance:** 3
**Recommendation:** 6

**Strength And Weaknesses:**

Strengths:

+ The method is simple and practical, with the additional benefit of avoiding the need to set aside clean label dataset.

+ The experiments are extensive, the results convincingly show that DisagreeNet performs well in general in regard to the state of the art. Some minor details such as confidence intervals in Fig 2 (a) would be welcome.

Weaknesses or questions:

- The main question is related to the novelty, as the use of the predictions of ensembles of models to filter noisy labeled data is not novel (See point 2 below).

- The theoretical claim of Section 3.1 does not bring much to the paper (and it weakens it in my opinion): first it applies only a linear regression with GD, which is a very easy setting to handle since it can be solved exactly, and second the proof is not clear despite the simplicity of the setting (see point 3 below).

- Despite its simplicity, the method can be computationally heavy because it can require some ten models (for instance in hard noise scenarios, as mentioned in Section 5.4). Although the experimental comparison is extensive on the performance of estimating label noise level, it does not include this metric.

**Summary Of The Paper:**

The paper observes experimentally that different deep network models learn datapoints at the different paces when trained with noisy labels, in contrast to the same learning dynamics known to hold for different deep network models when the labels are clean. It uses this observation to estimate the presence and level of label noise using a simple algorithm (DisagreeNet) that computes the difference between the training paces of different models via the ELP ("Ensemble Learning Pace") metric.

**Summary Of The Review:**

This is a paper over an interesting and timely topic, which proposes a solution that comes with practical advantages, such as its simplicity and the absence of the need for a clean fraction of the dataset or for priors on the noise level. Despite its simplicity, the method may be computationally heavy. A couple of similar schemes have been proposed earlier this year, and the novelty with respect to these should be clarified. The experiments should include them as benchmarks, but they are otherwise extensive and strong (contrary to the theoretical contribution).

---

### Official Review · Reviewer_gseX · 2022-10-24

**Confidence:** 4
**Correctness:** 4
**Technical Novelty And Significance:** 2
**Empirical Novelty And Significance:** 2
**Recommendation:** 5

**Clarity, Quality, Novelty And Reproducibility:**

I am split on this paper's novelty. In essence, this method is exactly the same as the cumulative training accuracy metric of Jiang et. al. 2021, except that it averages the metric over multiple runs. If viewed this way, the entire discussion about "agreement" between multiple models of an "ensemble" is irrelevant. There is only a pseudo-version of agreement which is actually cumulative accuracy at that epoch. There is no notion of an ensemble of predictors, but rather an independent set of random initializations robustify the metric.

However, on the other hand, even though this metric existed in past work, the authors do a good job at noise estimation, and demonstrating that the method holds promise across a wide range.

**Strength And Weaknesses:**

## Strengths
1. Paper is well-written and easy to follow, and the experiments and ablation studies are comprehensive.
2. I find the result on noise estimation by fitting a bimodal distribution very interesting. (Has this been used in prior work as well?)
3. The paper shows strong results in filtering out mislabeled examples, and estimating noise level. This helps achieve close to oracle generalization accuracy.

## Weaknesses
1. The theoretical argument requires all the models to overfit at the same iteration to cleanly observe the decrease in correlation. This appears to be a very strong assumption that will not hold in practice in an ensemble of models, especially because the goal is to enhance diversity between them.
2. Missing comparisons with state-of-the-art methods like ELR and SOP (https://proceedings.mlr.press/v162/liu22w/liu22w.pdf). We need to compare the numbers on the same training splits, hence, looking at table numbers will not be a fair comparison, but it appears that SOP outperforms the proposed method in some settings, especially in high noise or asymmetric settings.
3. The objective of the removal of examples seems sub-optimal. If you can identify the noisy examples, it might be helpful to train the model with their corrected label (this should be easy to estimate using training dynamics).
4. I am concerned about the novelty of the work. The ELP method when averaged over the entire training process is the same as the method by Jiang 2021 when averaged over multiple random runs of the same dataset (which in this case becomes the equivalent of the ensemble)

**Questions**
1. "Additionally, many of these methods work in an end-to-end manner, and thus neither provide noise level estimation nor do they deliver separate sets of clean and noisy data for novel future usages." What do you mean by this, and why can't the prior methods help clean data?

**Minor Comments**
1. Use \citep{} in 1st paragraph of the Introduction and \citet{} in last paragraph on Page 2.
2. We then show that this statistics -> statistic
3. Use `quotation' rather than 'quotation' for correct compilation.
4. Usage of overfit v/s overfitting
5. Please edit Figure 5 with smaller legend, and different limits for the third subfigure



**Summary Of The Paper:**

The paper shows that when training an ensemble of deep models with noisy data, different models learn different noisy data points at different times. However, as far as clean examples are considered, the training times are highly similar. The authors propose a method called ELP to predict which examples in a dataset are mislabeled and worth filtering out. The key idea is to calculate the average rate of training of an example across multiple models in an ensemble and use it as a robust method of estimating the learning time of the example. The authors show an interesting analysis of agreement between models and the existence of a bimodal trend of learning speed in Section 3.2 which I find to be the most interesting section of the paper.

The results on multiple datasets, noise types, and noise strengths are strong and perform competitively with the state of the art. Applications of the work include noise filtration and estimation, and training after dataset cleansing.




**Summary Of The Review:**

While the paper achieves strong results in removing mislabeled examples from a dataset, the method is very similar to a past method that was not as extensively evaluated. I am split on the significance of the paper.

---

### Official Review · Reviewer_d5tP · 2022-10-25

**Confidence:** 3
**Correctness:** 3
**Technical Novelty And Significance:** 3
**Empirical Novelty And Significance:** 2
**Recommendation:** 5

**Clarity, Quality, Novelty And Reproducibility:**

Clarity, Quality, Novelty And Reproducibility are provided in Strength and Weakness Section.

**Strength And Weaknesses:**

Strength
- This paper is well written and easy to follow.
- It seems to outperforms other competitors with large margin in various benchmark dataset.

Weakness
- For computing ELP score, the set of epochs \mathcal{\epsilon} could be prespecified. How did you choose it? I think it needs the validation set for determining it. In this respect, The argument that the author does not need a validation set may not be valid.
- What happens when an ensemble technique is applied to standard methods based on the memorization effct. My conjecture is that similar improvements could be achieved. Extensive comparisons should be done.
- The competitors are outdated. The recent works relevant with this paper are [1, 2, 3, 4].

[1] Yazhou Yao, Zeren Sun, Chuanyi Zhang, Fumin Shen, Qi Wu, Jian Zhang, Zhenmin Tang, Jo-SRC: A Contrastive Approach for Combating Noisy Labels, Proceedings of the IEEE/CVF Conference on Computer Vision and Pattern Recognition (CVPR), 2021, pp. 5192-5201.
[2] Taehyeon Kim, Jongwoo Ko, Sangwook Cho, Jinhwan Choi, Se-Young Yun, FINE Samples for Learning with Noisy Labels, In NeurIPS, 2021.
[3] Yingbin Bai*, Erkun Yang*, Bo Han, Yanhua Yang, Jiatong Li, Yinian Mao, Gang Niu, and Tongliang Liu, Understanding and Improving Early Stopping for Learning with Noisy Labels, In NeurIPS, 2021.
[4] Junnan Li, Richard Socher, Steven C.H. Hoi, DivideMix: Learning with Noisy Labels as Semi-supervised Learning, ICLR 2020.

**Summary Of The Paper:**

This paper observes that the dynamics of the ensemble model provide a distinction between clean and noisy labeled data in the training phase. The diversity of noisy labeled data is much larger than that of clean labeled data. Based on this observation, author proposes DisagreeNet which composes of the three steps : (1) noise rate estimation (2) noise filtration (3) supervised learning. Experiments show that DisagreeNet beats other competitors.

**Summary Of The Review:**

While the subject of this experimental paper is interesting and novel, I believe the authors' claims are not adequately supported due to a series of weaknesses of the experimental evaluation. I am more than willing to increase my score if the authors address the aforementioned limitations.

---

### Decision · Program_Chairs · 2023-01-20

**Decision:**

Reject

**Justification For Why Not Higher Score:**

According to my expertise and reviewing process, this paper should belong to a Reject.

**Justification For Why Not Lower Score:**

According to my expertise and reviewing process, this paper should belong to a Reject.

**Metareview: Summary, Strengths And Weaknesses:**

The paper studies the problem of label-noise learning, and proposes to compute a score based on the agreements over an ensemble of model, which identify the samples with the correct given labels. The paper conducts experiments and shows that the disagreements among the ensemble of models strongly correlate with the overfitting behavior, indicating that the labels could be noisy. The paper utilizes such an observation to identify samples with clean labels and use other label-noise learning or semi-supervised learning method to do the training. The strengths of this paper come from an interesting observation, which correlates the identification of noisy labels to the disagreements of an ensemble of models. Moreover, the paper proposes a very simple method to identify noisy-labeled samples based on observations about disagreements among an ensemble of models.

However, there are several obvious weakness: 1) The significance of novelty is limited. Particularly, as also mentioned by other reviewers, the idea of relying on consensus/confidence among two or multiple models to identify clean/noisy labels has been applied in the previous method. 2） The method is not end-to-end, and the comparison results against SOTA noisy learning method are weak. 3） It requires more computational costs for the proposed method as an ensemble of models are required. No comparison is given with methods using similar computational costs (e.g., larger models). Even though the authors argue that as few as 2-3 models can give good improvements, it could still be 2-3 times more costly compared to baselines. Therefore, a more careful comparison with baselines, including the computational costs, is necessary. Overall, this paper may not be ready for publication at ICLR. The next version must be a strong paper if authors can take comments into consideration.